# Belly roll, a GPI-anchored Ly6 protein, regulates *Drosophila melanogaster* escape behaviors by modulating the excitability of nociceptive peptidergic interneurons

Kai Li[1], Yuma Tsukasa[1], Misato Kurio[1], Kaho Maeta[2], Akimitsu Tsumadori[1], Shumpei Baba[1], Risa Nishimura[1], Akira Murakami[3], Koun Onodera[1], Takako Morimoto[4], Tadashi Uemura[1,5], Tadao Usui[1]*

[1]Graduate School of Biostudies, Kyoto University, Kyoto, Japan; [2]Faculty of Agriculture, Kyoto University, Kyoto, Japan; [3]Faculty of Science, Kyoto University, Kyoto, Japan; [4]School of Life Sciences, Tokyo University of Pharmacy and Life Sciences, Hachioji, Japan; [5]Research Center for Dynamic Living Systems, Kyoto University, Kyoto, Japan

*For correspondence: usui.tadao.3c@kyoto-u.ac.jp

**Competing interest:** The authors declare that no competing interests exist.

**Abstract** Appropriate modulation of escape behaviors in response to potentially damaging stimuli is essential for survival. Although nociceptive circuitry has been studied, it is poorly understood how genetic contexts affect relevant escape responses. Using an unbiased genome-wide association analysis, we identified an Ly6/α-neurotoxin family protein, Belly roll (Bero), which negatively regulates *Drosophila* nociceptive escape behavior. We show that Bero is expressed in abdominal leucokinin-producing neurons (ABLK neurons) and *bero* knockdown in ABLK neurons resulted in enhanced escape behavior. Furthermore, we demonstrated that ABLK neurons responded to activation of nociceptors and initiated the behavior. Notably, *bero* knockdown reduced persistent neuronal activity and increased evoked nociceptive responses in ABLK neurons. Our findings reveal that Bero modulates an escape response by regulating distinct neuronal activities in ABLK neurons.

## Editor's evaluation

This convincing and fundamental study uses unbiased genome-wide association analysis to identify a gene, called Bero, encoding Ly6/α-neurotoxin family protein, that affects the way larval *Drosophila* respond to nociceptive stimuli. This discovery is followed up by the identification of neurons in which Bero function is relevant for the modulation of nociceptive behaviour, namely the abdominal leucokinin-producing neurons. These neurons are activated by nociceptive sensory neurons and can initiate escape behavior. In these neurons Bero modulates both persistent and evoked activities. This elegant work will be of interest to neurobiologists working on genes, neural circuits, and behaviour.

## Introduction

Appropriate escape behavior in response to threatening stimuli is essential for organismal survival (*Branco and Redgrave, 2020*; *Burrell, 2017*; *Chin and Tracey, 2017*; *Im and Galko, 2012*; *Peirs and Seal, 2016*). In both vertebrates and invertebrates, the modulation of escape behavior depends on environmental contexts as well as intensity and modality of stimuli, which are sensed and processed

in a multilayered neuronal network, subject to genetic variation, that integrates external and internal sensory stimuli (*Burnett et al., 2016*; *Jennings et al., 2014*; *Mu et al., 2012*; *Ohyama et al., 2015*). Within the network, peptidergic neuromodulation plays a critical role in neuronal information processing (*Nässel and Winther, 2010*; *Taghert and Nitabach, 2012*).

When *Drosophila melanogaster* larvae encounter noxious stimuli, such as harsh mechanical stimulations intrinsic to attacks by parasitoid wasps, they show stereotypic escape behaviors composed of an initial abrupt bending and subsequent corkscrew-like rolling (*Hwang et al., 2007*; *Onodera et al., 2017*; *Tracey et al., 2003*). In larvae, the noxious stimuli are mainly sensed by class IV dendritic arborization neurons (Class IV neurons) underneath the body wall (*Tracey et al., 2003*). In addition, a type of tracheal dendrite neuron (v'td2 neurons) and class III dendritic arborization neurons (Class III neurons) are known to detect noxious light stimuli and noxious cold stimuli, respectively (*Imambocus et al., 2022*; *Turner et al., 2016*). Recent studies have identified several important downstream components of the nociceptive neural circuitry (*Ohyama et al., 2015*; *Takagi et al., 2017*; *Yoshino et al., 2017*). In parallel, several genes and neurotransmitters have been characterized as context-dependent modulators of the behaviors (*Burgos et al., 2018*; *Dason et al., 2020*; *Hu et al., 2017*; *Kaneko et al., 2017*). While most of these studies identified the nociceptive neural circuitry by connectomic approaches, we used a genetic strategy to provide additional insights into the nociceptive modulation. Importantly, we discovered that *D. melanogaster* larvae displayed an extensive natural variation in nociceptive escape behavior. This broad behavioral variation allowed us to perform an unbiased genome-wide association (GWA) analysis to explore genetic variants affecting the nociceptive escape behaviors.

In this study, we have demonstrated that an Ly6/α-neurotoxin family protein, Belly roll (Bero), modulates the neuronal excitability of peptidergic interneurons that initiates and facilitates *Drosophila* nociceptive escape behaviors. We first identified the *bero* gene through GWA analysis and validated its role in regulating nociception using pan-neuronal *bero* knockdown and null mutant animals. By searching for *bero*-expressing neurons, we identified abdominal leucokinin-producing neurons (ABLK neurons) that have been reported to be involved in controlling larval escape behaviors (*Hu et al., 2020*; *Imambocus et al., 2022*). We then found that *bero* knockdown in ABLK neurons resulted in enhanced escape behaviors, similar to *bero* mutants. Furthermore, we demonstrated that ABLK neurons responded to the activation of nociceptors and initiated nociceptive escape behavior. Notably, *bero* knockdown in ABLK neurons resulted in reduced persistent activities and increased evoked nociceptive responses in ABLK neurons, implying an essential role of *bero* in regulating neuronal activities. Based on our results regarding the ABLK neurons, we propose that these neurons may integrate external and internal sensory stimuli, thereby orchestrating nociceptive escape behaviors.

## Results

### Natural variation in nociceptive responses in wild-type strains of the *Drosophila melanogaster* Genetic Reference Panel

Upon noxious thermal stimulation, *Drosophila* larvae show stereotypical nociceptive responses: bending and rolling behavior (*Figure 1A*; *Tracey et al., 2003*). Using the Heat Probe Assay (HPA; *Tracey et al., 2003*), we noticed that there is a large difference in the latency of rolling behavior (rolling latency) between the two commonly used strains, $w^{1118}$ and *Canton-S* (*Figure 1B*). In order to evaluate natural variation in nociceptive responses, we took advantage of the *Drosophila melanogaster* Genetic Reference Panel (DGRP; *Huang et al., 2014*; *Mackay et al., 2012*), a collection of wild-caught *Drosophila* lines. We quantified the rolling behavior in 38 representative inbred wild-type strains of DGRP and observed a considerable variation in the behavior: the strain-specific median rolling latency ranged from 2.83 s to 10 s and the occurrence rates of rolling behavior (rolling probability) within 10 s ranged from around 30% to almost 100% among these strains (*Figure 1C and D*). We categorized the rolling probability into three response classes (rolling probability in 2, 5, and 10 s) based on previous studies (*Onodera et al., 2017*; *Terada et al., 2016*; *Tracey et al., 2003*) and found that the three response classes are highly correlated with one another (*Figure 1—figure supplement 1A*). To identify genetic variants associated with the behavioral variation, we performed multiple trials of GWA mapping analyses using the publicly available web-based analysis tool, DGRP 2.0 (*Huang et al., 2014*), with four statistical metrics that characterize larval rolling behavior and are mutually dependent to some extent (*Figure 1E*, *Figure 1—source data 1*, 'Materials and methods'). We then

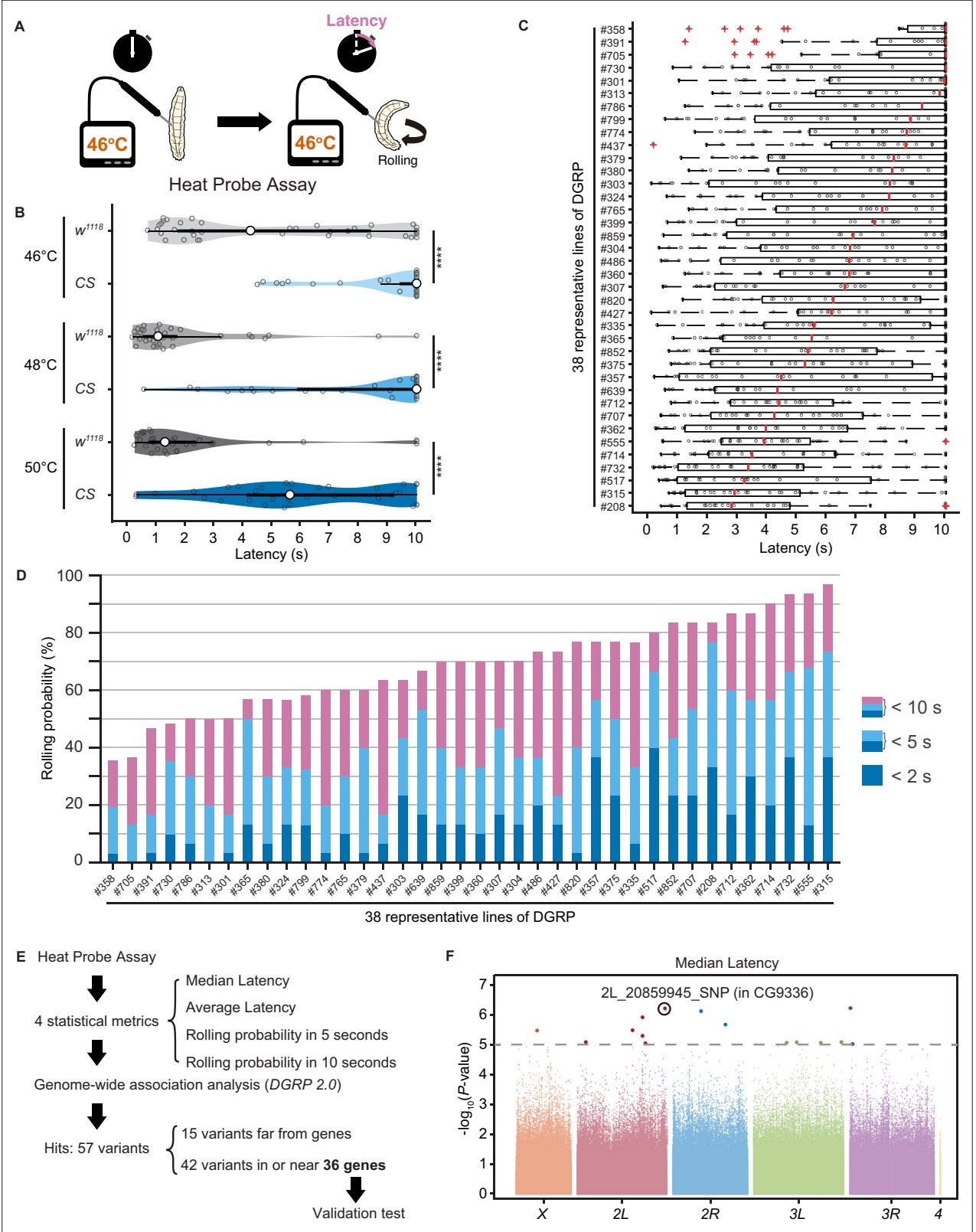

**Figure 1.** Natural diversity in nociceptive rolling escape behavior in wild-type strains of *Drosophila melanogaster*. (**A**) A schematic representation of the Heat Probe Assay (see 'Materials and methods' for details). (**B**) The latency of rolling escape behavior (rolling latency) is different between two wild-type strains, $w^{1118}$ and *Canton-S* (*CS*) (46°C, [$w^{1118}$] n = 40, [*CS*] n = 38; 48°C, [$w^{1118}$] n = 32, [*CS*] n = 33; 50°C, [$w^{1118}$] n = 33, [*CS*] n = 33; ***p<0.001, Wilcoxon rank-sum test). Violin plots provide a kernel density estimate of the data, where the middle circle shows the median, a boxplot shape indicates 25th

*Figure 1 continued on next page*

*Figure 1 continued*

and 75th percentiles, and whiskers to the left and right of the box indicate the 90th and 10th percentiles, respectively, in this and the following figures. Each data point represents an individual larva. See source data tables for detailed genotypes in this and the following figures. (**C**) The rolling latency of 38 representative *Drosophila melanogaster* Genetic Reference Panel (DGRP) lines in ascending order (n = 30 or 31 larvae/line). Boxplots indicate the median and 25th and 75th percentiles, and whiskers to the left and right of the box indicate the 90th and 10th percentiles, respectively, in this and the following figures. (**D**) The rolling probability of 38 representative DGRP lines in ascending order (n = 30 or 31 larvae/line). The stacked bar chart indicates the rolling probability within 2, 5, and 10 s, respectively. (**E**) A diagram showing experimental procedures for the genome-wide association (GWA) analysis. (**F**) GWA analysis for median rolling latency of the rolling escape behavior. The p-values ($-\log_{10}$ transformed) are shown on the *y* axis. The gray dotted line marks the nominal p-value threshold ($1.0 \times 10^{-5}$). Each data point corresponds to an individual genetic variant (single-nucleotide polymorphism, deletion, or insertion). Data points are arranged by relative chromosome (genomic) position, the color code indicates the respective chromosome to which they belong. See also *Figure 1—source data 1* and *Figure 1—figure supplement 1A*.

The online version of this article includes the following source data and figure supplement(s) for figure 1:

**Source data 1.** Summary table of genotypes, statistical testing, and graph data for *Figure 1*.

**Source data 2.** Genome-wide association results for rolling behavior.

**Figure supplement 1.** Correlation analysis with rolling probability in three response classes and genome-wide association (GWA) analysis.

**Figure supplement 1—source data 1.** 4 Statistical metrics of rolling behavior of *Drosophila melanogaster* Genetic Reference Panel (DGRP) lines.

**Figure supplement 1—source data 2.** Summary table of genotypes, statistical testing, and graph data for *Figure 1—figure supplement 1*.

identified in total 37 single-nucleotide polymorphisms (SNPs), two deletions, and three insertions in or near 36 candidate genes (*Figure 1—source data 2*), at a threshold p-value of $1.0 \times 10^{-5}$, a commonly used threshold p-value in previous studies with DGRP (*Figure 1F*, *Figure 1—figure supplement 1B*; *Mackay and Huang, 2018*).

## Identification of *CG9336/belly roll* (*bero*) through an RNA interference knockdown screen of candidate genes

To explore impacts of these candidate genes on nociceptive responses, we carried out a secondary functional screen by pan-neuronal RNA interference (RNAi)-mediated gene knockdown. By evaluating 17 RNAi strains that were currently available, we found that pan-neuronal knockdown of the *CG9336* gene resulted in significant enhancement of the rolling probability (*Figure 2A*). To further test whether the phenotype was specifically caused by suppression of *CG9336*, we generated two distinct short-hairpin RNAs (shRNAs) against *CG9336* (shRNA#1 and #2; *Figure 2B*; 'Materials and methods'), validated the efficiency of shRNA#2 (*Figure 2—figure supplement 1A and B*; 'Materials and methods'), and found that both shRNAs caused a comparable enhancement (*Figure 2E*, *Figure 2—figure supplement 1C*). We also performed three sets of effector-control tests for *UAS-bero shRNA#2* with different genomic backgrounds. In each group, both the *UAS-bero shRNA#2* control animal and the corresponding controls exhibited comparable rolling behavior, which ensures that any observed phenotypes were not due to leaky expression from the UAS transgene (*Figure 2E*). We then designated the gene as *belly roll* (*bero*), after a high-jump style in athletics. *bero* encodes a member of the Ly6/α-neurotoxin protein superfamily, which is characterized by a three-finger structure, including various membrane-tethered and secreted polypeptides (*Loughner et al., 2016*). Using several bioinformatics tools, we predicted a signal peptide sequence at the amino terminus, two possible glycosylation sites, a GPI-anchorage site, a putative transmembrane region at the carboxyl terminus, and five putative disulfide bridges, which are typical structural features of the Ly6/α-neurotoxin protein superfamily (*Figure 2C, Figure 2—figure supplement 1D*; *Almagro Armenteros et al., 2019*; *Baek et al., 2021*; *Eisenhaber et al., 1999*; *Gupta and Brunak, 2001*; *Krogh et al., 2001*). A three-dimensional model of Bero was generated by protein structure prediction using AlphaFold2 (*Jumper et al., 2021*; *Figure 2D*).

The *bero*-associated SNP, 2L_20859945_SNP, is located in the second intron of *CG9336 isoform A* (FBtr0081427) and comprises two nucleotide variations in the DGRP core 40 lines: a minor allele, cytosine, and a major allele, thymine (*Figure 2B*). We found that most of the 38 DGRP core lines with the *bero* minor allele (C) showed higher responsiveness than those with the major allele (T) upon noxious thermal stimulations (*Figure 2—figure supplement 2A*). Based on these observations and our *bero* knockdown results, we speculated that animals with the *bero* minor allele (C) might have lower expression levels of *bero* compared with those possessing the major allele (T). Thus, using

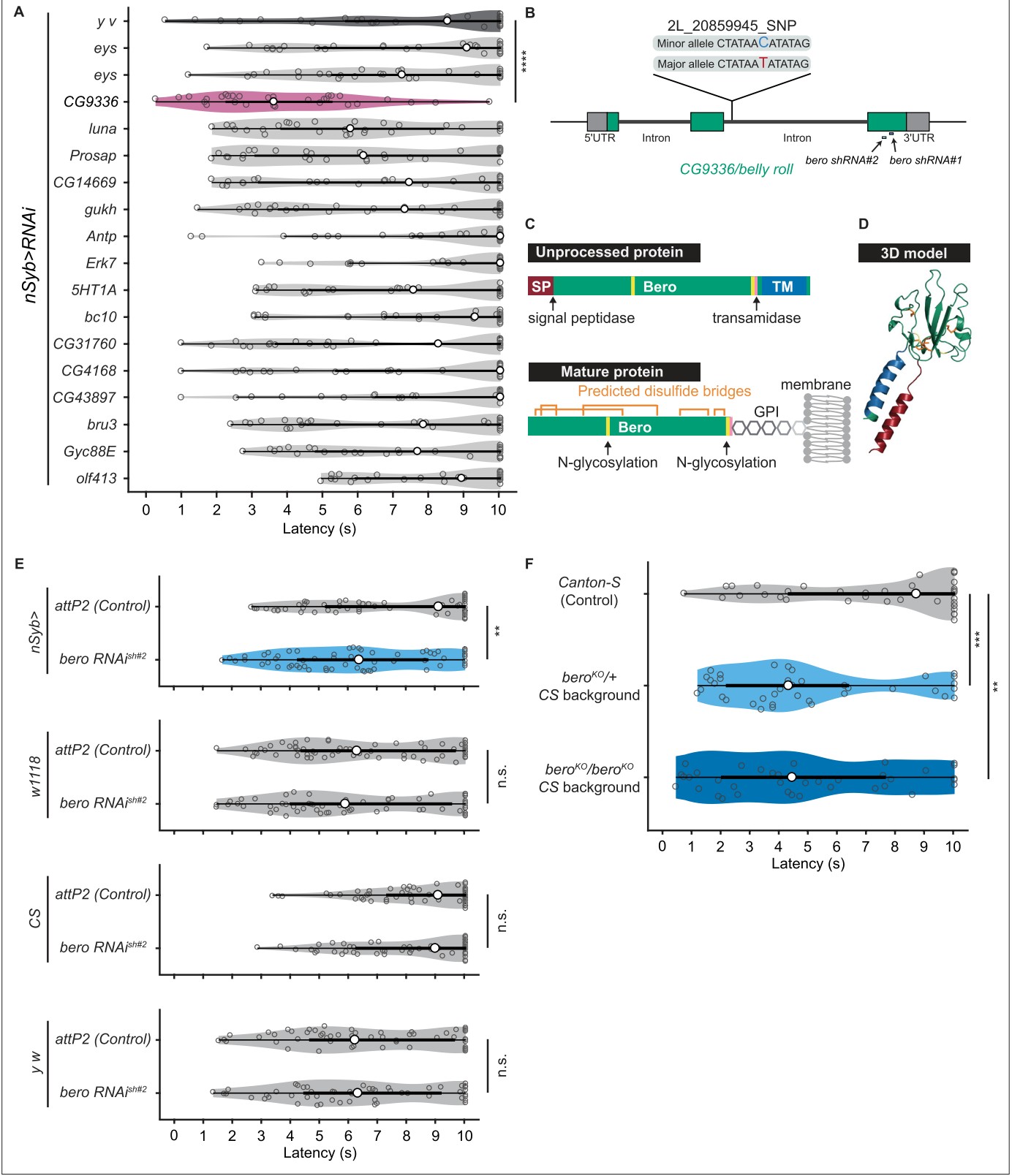

**Figure 2.** *bero* negatively regulates nociceptive rolling escape behavior. (**A**) A secondary functional screen by pan-neuronal RNA interference. The rolling latency of each UAS-RNAi line was measured. Pan-neuronal knockdown of *CG9336/bero* reduces rolling latency (Heat Probe Assay, n = 30 or 31 larvae/genotype; ****p<0.0001, Wilcoxon rank-sum test). (**B**) A schematic representation of *bero*-associated SNP (2L_20859945_SNP; minor allele, cytosine; major allele, thymine; minor allele frequency = 0.4167). (**C**) A schematic representation of unprocessed and mature Bero protein. SP, signal

*Figure 2 continued on next page*

*Figure 2 continued*

peptide; TM, transmembrane region. Orange lines mark the predicted disulfide bonds. Yellow lines mark the predicted N-glycosylation sites. Pink lines mark the predicted GPI-modification site. See also *Figure 2—figure supplement 1D*. (**D**) A three-dimensional protein structure prediction of unprocessed Bero protein by AlphaFold2. Red region, signal peptide; blue region, transmembrane region; orange sticks, predicted disulfide bonds; yellow region, predicted N-glycosylation sites; pink region, predicted GPI-modification site. (**E**) Rolling latency of *nSyb>attP2* control (n = 63), pan-neuronal *bero* knockdown animals (*nSyb>bero RNAi^shRNA#2^*, n = 65), and the corresponding three sets of effector-controls (*w1118* background: *attP2* control, n = 64, *bero RNAi^shRNA#2^*, n = 60; *Canton-S* background: *attP2* control, n = 60, *bero RNAi^shRNA#2^*, n = 60, *y w* background: *attP2* control, n = 55, *bero RNAi^shRNA#2^*, n = 55). **p<0.01; n.s., nonsignificant; Wilcoxon rank-sum test. The genetic backgrounds are controlled (see source data tables for detailed genotypes in this and the following figures). (**F**) Rolling latency of *Canton-S* control (n = 38), *bero* heterozygous (*bero^KO^/+*, n = 38), and homozygous (*bero^KO^/bero^KO^*, n = 37) mutant animals. **p<0.01, ***p<0.001, Wilcoxon rank-sum test. The *bero* KO strain had been outcrossed to *Canton-S* for 11 generations.

The online version of this article includes the following source data and figure supplement(s) for figure 2:

**Source data 1.** Summary table of genotypes, statistical testing, and graph data for *Figure 2*.

**Figure supplement 1.** Validation of bero RNAishRNA#2, amino acid sequences of Bero protein, and generation of bero knockout strains.

**Figure supplement 1—source data 1.** Summary table of genotypes, statistical testing, and graph data for *Figure 2—figure supplement 1*.

**Figure supplement 2.** Quantification of *bero* gene expression.

**Figure supplement 2—source data 1.** Summary table of genotypes and graph data for *Figure 2—figure supplement 2*.

**Figure supplement 2—source data 2.** Full raw unedited gel and labeled figure with the uncropped gel for *Figure 2—figure supplement 2*.

reverse transcriptase PCR, we compared the expression levels of *bero* in the larval central nervous system (CNS) among four DGPR strains that showed extremely quick or slow behavioral responses. We found that animals with the minor allele have lower expression levels of *bero* (*Figure 2—figure supplement 2B*). Using online sequence analysis based on JASPAR 2022 (*Castro-Mondragon et al., 2022*), we predicted that the neighboring nucleotide sequences of the *bero*-associated SNP might be a target site of certain transcription factors that control the expression level of *bero* in CNS (*Supplementary file 2*). Furthermore, we generated a *bero* null mutant line using CRISPR-mediated mutagenesis, which deleted the entire protein-coding sequence (*Figure 2—figure supplement 1E*; 'Materials and methods'), and observed in HPA experiments that both *bero* heterozygous (*bero^KO^/+*) and homozygous (*bero^KO^/bero^KO^*) mutant animals displayed an enhanced rolling escape behavior (*Figure 2F*). Taken together, these results strongly suggest that *bero* expression in neurons negatively regulates nociceptive escape behavior.

## *bero* is expressed in several subgroups of peptidergic neurons in the larval CNS

We next sought to find which neurons express *bero* in the larval CNS. To do this, we examined the overlap between *bero*-expressing neurons labeled in a protein-trap strain *bero*-YFP (CPTI-001654; *Lowe et al., 2014*), in which the endogenous Bero protein is fused with a yellow fluorescent protein variant mVenus, and candidate neurons labeled by a CD4:tdTomato fusion protein driven by distinct GAL4-drivers. We found no *bero*-YFP expression in the Class IV neurons (nociceptors) or in the downstream neurons that have been reported to be required for the nociceptive escape behavior (*Figure 3—figure supplement 1A*; *Ohyama et al., 2015*). Consistent with this, *bero* knockdown specifically in the corresponding neurons did not alter the rolling behavior (*Figure 3—figure supplement 1B*). Instead, we observed *bero*-YFP expression in several groups of peptidergic neurons specified by the bHLH transcription factor *dimmed* (*dimm*), including insulin-producing cells (IPCs) and a subset of Eclosion hormone-producing neurons (EH neurons) in the larval brain, and abdominal leucokinin-producing neurons (ABLK neurons) in the lateral region of abdominal ganglions in the larval ventral nerve cord (VNC; *Figure 3A and B*). In support of our observations, a recent single-cell transcriptomic atlas showed that *bero/CG9336* is selectively enriched in peptidergic neurons (*Corrales et al., 2022*). Bero is also expressed in certain non-neuronal tissues, such as midline glia (*Figure 3A*). However, we did not study Bero function in the glia in this study.

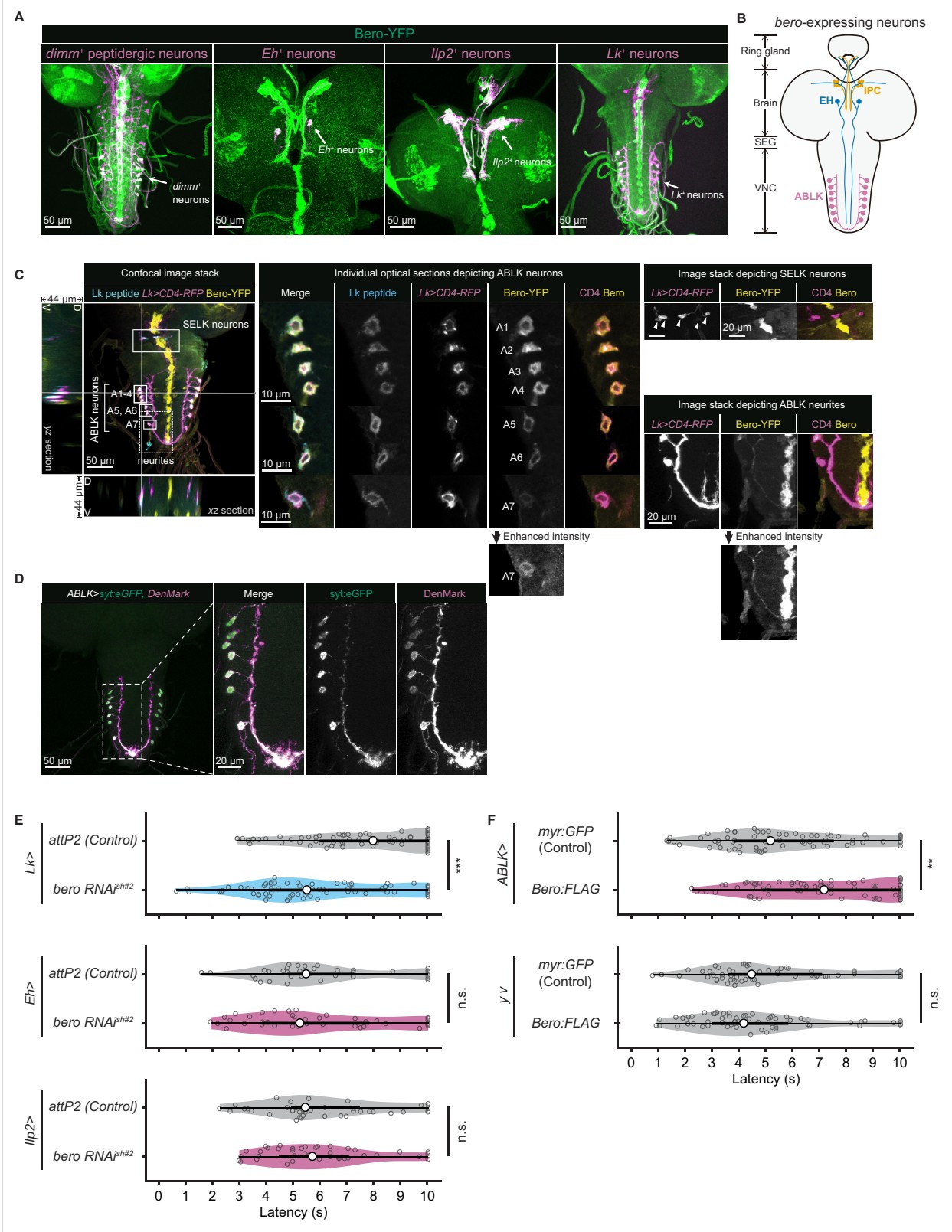

**Figure 3.** Expression of *bero* in ABLK neurons plays an essential role in the negative regulation of nociceptive behavior. (**A**) Confocal image stacks (maximum projection) showing fluorescence of the endogenous Bero reporter, Bero-YFP (green), and the colocalized peptidergic neurons with *GAL4*, *UAS-CD4-tdTomato* (magenta) in third-instar larvae. Scale bars, 50 μm. (**B**) A schematic representation of *bero*-expressing neurons. IPC, insulin-producing cells; EH, eclosion hormone-producing neurons; ABLK, abdominal leucokinin-producing neurons; SEG, the subesophageal ganglion; VNC,

*Figure 3 continued on next page*

Figure 3 continued

the ventral nerve cord. (C) Confocal image stack showing endogenous Bero expression (labeled by Bero-YFP, yellow), LK neurons (*Lk-GAL4.TH, UAS-CD4-tdTomato*, magenta), and leucokinin (labeled by anti-Lk, cyan) in a third-instar larva. Bottom and left panels show an *XZ* and *YZ* cross-section of the ABLK somatic region (locations of the cross-sections are indicated by horizontal and vertical gray lines in the primary image), respectively. Middle panels show magnified views of the boxed regions: Individual optical sections depicting ABLK neurons (ABLK neurons are indicated by A1–A7 label, and an image of A7 with enhanced intensity is showed), Right: image stack depicting SELK neurons (indicated by arrowheads), and image stack depicting ABLK neurites (an image with enhanced intensity is showed). Scale bars: 50 µm; 10 µm or 20 µm for the magnified view. (D) Confocal image stack showing the dendrite (labeled by *ABLK >DenMark,* magenta) and axon terminal markers (labeled by *ABLK >syt:eGFP*, green) in ABLK neurons in a third-instar larva. Scale bars: 50 µm; 20 µm for the magnified view. (E) Rolling latency of LK-specific *bero* knockdown larvae (*Lk>bero RNAi^shRNA #2*, n = 73) and control (*Lk>attP2,* n = 75). Rolling latency of Eh-specific *bero* knockdown larvae (*Eh>bero RNAi^shRNA #2*, n = 38) and control (*Eh>attP2,* n = 41). Rolling latency of Ilp2-specific *bero* knockdown larvae (*Ilp2>bero RNAi^shRNA #2*, n = 40) and control (*Ilp2>attP2,* n = 39). ***p<0.001; n.s., nonsignificant; Wilcoxon rank-sum test. (F) Rolling latency of ABLK-specific *bero* overexpression larvae (*ABLK>Bero:FLAG*, n = 72), control (*ABLK>myr:GFP*, n = 73), and the corresponding effector-controls (*y v* background: *myr:GFP* control, n = 60, *Bero:FLAG*, n = 61). **p<0.01; n.s., nonsignificant; Wilcoxon rank-sum test.

The online version of this article includes the following source data and figure supplement(s) for figure 3:

**Source data 1.** Summary table of genotypes, statistical testing, and graph data for *Figure 3*.

**Figure supplement 1.** *bero* knockdown in nociception-related neurons and specific overexpression of Bero:FLAG in ABLK neurons.

**Figure supplement 1—source data 1.** Summary table of genotypes, statistical testing, and graph data for *Figure 3—figure supplement 1*.

## *bero* expression in ABLK neurons negatively regulates nociceptive rolling escape behavior

To identify the particular subgroups of neurons that require *bero* expression for negative regulation of nociceptive escape behavior, we examined rolling escape behavior in larvae with cell type-specific RNAi knockdowns of *bero*. We found that knockdown of *bero* using *Lk-GAL4*, a leucokinin-producing neuron-specific driver, caused enhanced rolling escape behavior (*Figure 3E*). In contrast, knockdown of *bero* using either *Eh-GAL4*, a EH neuron-specific driver, or *Ilp2-GAL4*, an IPC-specific driver, did not result in any changes in rolling escape behavior (*Figure 3E*). LK neurons can be classified into four anatomically distinct subgroups in the larval CNS: anterior LK neurons (ALKs), lateral horn LK neurons (LHLKs), subesophageal LK neurons (SELKs), and abdominal LK neurons (ABLKs; *de Haro et al., 2010*). *bero* is expressed in all seven pairs of ABLK neurons and is delivered to their neurites, but not expressed in any other LK neurons (*Figure 3A and C*). Consistent with this, the seven pairs of *bero*-YFP labeled neurons were immunoreactive for Leucokinin (*Figure 3C*). High-resolution imaging of individual optical sections showed that Bero localizes on both the plasma membrane and internal membrane of ABLK neurons (*Figure 3C*, middle). We expressed the dendrite and axon terminal markers, namely DenMark and synaptotagmin-eGFP (syt:eGFP), in ABLK neurons (*Figure 3D*), demonstrating that Bero distributes along both dendric and axonal regions of the neurons. To investigate the role of *bero* expression in ABLK neurons in modulating rolling behavior, we tried to induce ABLK-specific *bero* overexpression using an intersectional flip-out approach (*Bohm et al., 2010*; *Simpson, 2016*; *Figure 3—figure supplement 1C*). We observed that *bero* overexpression suppressed rolling behavior (*Figure 3F*), indicating that *bero* expression in ABLK neurons plays an essential role in the negative regulation of nociceptive behavior. In contrast, in the effector-control test, both the *UAS-Bero:FLAG* control animal and the corresponding controls showed comparable levels of rolling behavior (*Figure 3F*). Hence, we focused our analysis on the physiological properties of ABLK neurons and the functional roles of *bero* in these neurons.

## *bero* promotes persistent fluctuating activities and inhibits evoked nociceptive responses in ABLK neurons

We next addressed whether ABLK neurons respond to stimulation of nociceptors. To test this, we expressed a blue light-gated cation channel ChR2.T159C (a variant of channelrhodopsin-2; *Berndt et al., 2011*) in nociceptors, and a red fluorescence $Ca^{2+}$ indicator jRCaMP1b (*Dana et al., 2016*) in LK neurons. We monitored calcium responses in ABLK neurons in fillet preparations and observed that cell bodies and neurite regions of ABLK neurons displayed persistent fluctuations of intracellular $Ca^{2+}$ concentration (*Figure 4A and B*, *Figure 4—figure supplement 1A*), which was consistent with a previous study (*Okusawa et al., 2014*). Notably, optogenetic activation of the nociceptors evoked weak but reliable evoked neuronal activities in ABLK neurons, suggesting that these neurons respond

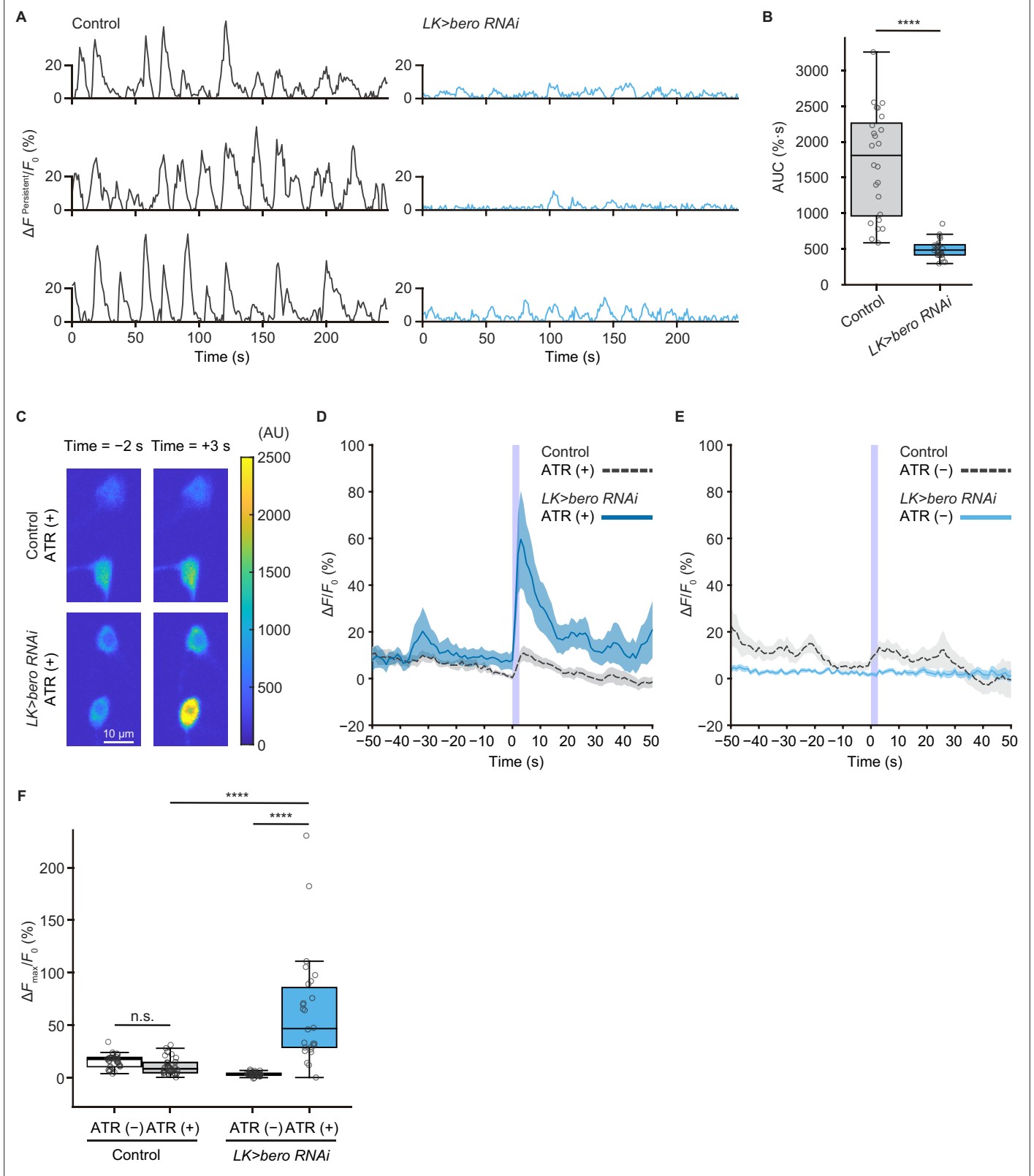

**Figure 4.** *bero* is necessary for the maintenance of persistent fluctuating activities and suppression of acute evoked nociceptive activity in ABLK neurons. (**A**) Representative results of calcium responses showing the persistent fluctuating calcium signal ($\Delta F^{\text{Persistent}}/F_0$) of ABLK neurons in control (*TrpA1>ChR2.T159C, LK>jRCaMP1b*) and *bero* knockdown larvae (*TrpA1>ChR2.T159C, LK>jRCaMP1b, bero RNAi^{shRNA #2}*). (**B**) A quantitative comparison of area under the curve (AUC) of persistent fluctuating activities of ABLK neurons in control (n = 24 neurons from three animals) and *bero* knockdown

*Figure 4 continued on next page*

*Figure 4 continued*

larvae (n = 25 neurons from three animals). ****p<0.0001, Wilcoxon rank-sum test. (**C**) Representative confocal images of the acute nociceptive responses of ABLK neurons in control (*TrpA1>ChR2.T159C, LK>jRCaMP1b*) and *bero* knockdown larvae (*TrpA1>ChR2.T159C, LK>jRCaMP1b, bero RNAi^shRNA #2^*) after ChR2.T159C-mediated optogenetic activation of Class IV neurons (nociceptors). Scale bars, 10 μm. (**D**) Time series for calcium responses ($\Delta F/F_0$; see 'Materials and methods' for details) of ABLK neurons in control (*TrpA1>ChR2.T159C, LK>jRCaMP1b*, ATR (+), n = 29 neurons from three animals) and *bero* knockdown larvae (*TrpA1>ChR2.T159C, LK>jRCaMP1b, bero RNAi^shRNA #2^*, ATR (+), n = 27 neurons from three animals) upon optogenetic activation of Class IV neurons (nociceptors). Blue light (470 nm) application is indicated by violet shading beginning at Time 0, and continued for 2.5 s. Light blue and gray shading indicate ± SEM. (**E**) Time series for calcium responses of ABLK neurons in control (*TrpA1>ChR2.T159C, LK>jRCaMP1b*, ATR (−), n = 24 neurons from three animals) and *bero* knockdown larvae (*TrpA1>ChR2.T159C, LK>jRCaMP1b, bero RNAi^shRNA #2^*, ATR (−), n = 25 neurons from three animals) upon optogenetic activation of Class IV neurons. (**F**) Quantitative comparison of maximum calcium responses ($\Delta F_{max}/F_0$) of ABLK neurons in control (ATR (−), n = 24 neurons from three animals; ATR (+), n = 29 neurons from three animals) and *bero* knockdown larvae (ATR (−), n = 25 neurons from three animals; ATR (+), n = 27 neurons from three animals) upon optogenetic activation of Class IV neurons (nociceptors). ****p<0.0001; n.s., nonsignificant; Kruskal−Wallis test followed by Dunn's test.

The online version of this article includes the following source data and figure supplement(s) for figure 4:

**Source data 1.** Summary table of genotypes, statistical testing, and graph data for *Figure 4*.

**Figure supplement 1.** Persistent fluctuating activities at the neurites of ABLK neurons and calcium responses recording traces of control and *bero* knockdown larvae.

**Figure supplement 1—source data 1.** Summary table of genotypes for *Figure 4—figure supplement 1*.

**Figure supplement 2.** Knockdown of *bero* in LK neurons does not affect the free locomotion of larvae.

**Figure supplement 2—source data 1.** Summary table of genotypes, statistical testing, and graph data for *Figure 4—figure supplement 2*.

to nociceptive inputs (*Figure 4C and D*; *Hu et al., 2020*). We used larvae reared on a medium with no additional all *trans*-retinal (ATR) as controls since optogenetic activation requires extra ATR in the food. A $Ca^{2+}$ rise was also detected in the control larvae reared on a medium with no additional ATR, but the rising phase preceded the onset of the blue light illumination (*Figure 4E*). This indicates that the $Ca^{2+}$ rise could be attributed to the persistent fluctuation in ABLK neurons.

To further investigate the physiological roles of *bero* in ABLK neurons, we tested whether knockdown of *bero* alters persistent activities and/or nociceptive responses of the neurons. We found that *bero* knockdown almost completely abolished persistent fluctuating activities (*Figure 4A and B*), whereas it increased the amplitude of evoked nociceptive responses (*Figure 4C, D and F*). Given that *bero* knockdown enhanced rolling escape behavior in the HPA experiments (*Figure 3E*), we speculated that the evoked nociceptive responses in ABLK neurons are important for facilitating rolling escape behavior. Given the phenotype of the *bero* knockdown on spontaneous persistent activities in ABLK neurons, we performed a behavioral analysis of undisturbed free locomotion and found that *bero* knockdown did not affect larval free locomotion (*Figure 4—figure supplement 2A–C*).

Remarkably, under ATR (+) condition, persistent fluctuations in ABLK neurons of *bero* knockdown animals appeared to be enhanced compared to controls (*Figure 4D*, *Figure 4—figure supplement 1B*). We postulate that the 561 nm laser used for jRCaMP1b excitation weakly activates ChR2.T159C in nociceptors, inducing weak nociceptive activity. Furthermore, ABLK neurons with *bero* knockdown displayed increased sensitivity to these nociceptive inputs than controls, which may provide a possible explanation. Therefore, the baseline neural activity under ATR (+) conditions may not represent proper spontaneous activity. Thus, we only compared spontaneous activity under the ATR (−) condition.

Collectively, we have revealed that *bero* plays a critical role in not only generating a relatively higher level of persistent fluctuating activities, but also inhibiting the evoked nociceptive responses of the ABLK neurons, thereby suppressing the nociceptive escape behavior (Figure 6C).

## Proper dynamics of acute evoked activity in ABLK neurons is necessary for the control of rolling escape behavior

To test whether direct manipulations of ABLK activity lead to escape behaviors, we optogenetically activated the neurons using a red light-gated cation channel CsChrimson (*Klapoetke et al., 2014*). We first activated pan-LK neurons and found that red light illumination elicited a robust rolling behavior that was never observed in control larvae (*Figure 5A and B*). We then employed the intersectional flip-out approach to selectively express CsChrimson in ABLK or SELK neurons (*Figure 5B*). Surprisingly, optogenetic activation of ABLK neurons almost exclusively resulted in not rolling but bending behavior, which is considered as a separable initial phase of the entire sequence of escape motions

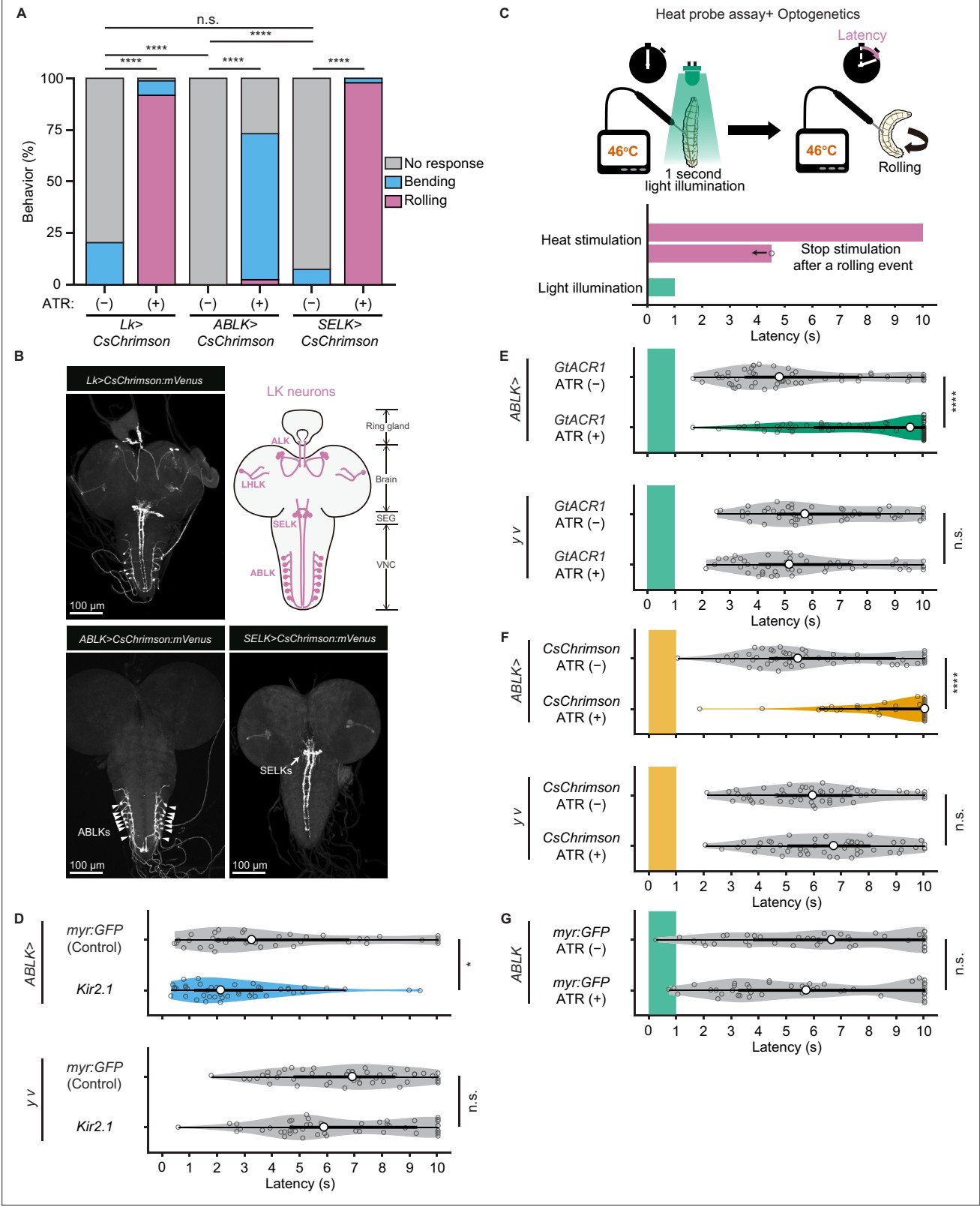

**Figure 5.** Proper dynamics of acute activity in ABLK neurons is necessary for the control of rolling escape behavior. (**A**) Percentage of larvae showing nociceptive escape behavior (rolling and bending) upon optogenetic activation of LK neurons (*Lk>CsChrimson*, ATR (+), n = 85; ATR (−), n = 49), ABLK neurons (*ABLK>CsChrimson*, ATR (+), n = 41; ATR (−), n = 22), and SELK neurons (*SELK>CsChrimson*, ATR (+), n = 46; ATR (−), n = 54). ****p<0.0001; n.s., nonsignificant; Chi-squared test. (**B**) Confocal image stack showing the labeling of all LK neurons (*Lk>CsChrimson.mVenus*), ABLK neurons

*Figure 5 continued on next page*

*Figure 5 continued*

(*ABLK>CsChrimson.mVenus*), and SELK neurons (*SELK>CsChrimson.mVenus*) in third-instar larvae. Scale bars: 100 μm. Top-right panel shows a schematic map of LK neuron types. (**C**) A schematic representation of the Heat Probe Assay combined with optogenetic manipulation (see 'Materials and methods' for details). The heat stimulation and light illumination were delivered almost simultaneously: the larva was touched laterally with a heat probe until the initiation of the first 360° rotation of the animal. The response latency was computed as 10 s if it was more than 10 s. A 1-s-long light pulse was delivered for optogenetic manipulation. (**D**) Rolling latency of ABLK-specific inhibition larvae (*ABLK>Kir2.1*, n = 45), control larvae (*ABLK>myr:GFP*, n = 42), and the corresponding effector-controls (*y v* background: *myr:GFP* control, n = 50, *Kir2.1*, n = 50) in the Heat Probe Assay. *p<0.05; n.s., nonsignificant; Wilcoxon rank-sum test. (**E**) Rolling latency of ABLK-specific optogenetic inhibition larvae (*ABLK>GtACR1*, ATR (+), n = 68), control larvae (*ABLK>GtACR1*, ATR (−), n = 54), and the corresponding effector-controls (*y v* background: *GtACR1*, ATR (−), n = 50, *GtACR1*, ATR (+), n = 50) in the Heat Probe Assay. ****p<0.0001; n.s., nonsignificant; Wilcoxon rank-sum test. (**F**) Rolling latency of ABLK-specific optogenetic activation larvae (*ABLK>CsChrimson*, ATR (+), n = 53), control larvae (*ABLK>CsChrimson*, ATR (−), n = 52), and the corresponding effector-controls (*y v* background: *CsChrimson*, ATR (−), n = 50, *CsChrimson*, ATR (+), n = 50) in the Heat Probe Assay. ****p<0.0001; n.s., nonsignificant; Wilcoxon rank-sum test. (**G**) Rolling latency of the corresponding driver-control larvae (*ABLK >myr:GFP*, ATR (+), n = 46, *ABLK>myr:GFP*, ATR (−), n = 45) in the Heat Probe Assay. n.s., nonsignificant; Wilcoxon rank-sum test.

The online version of this article includes the following source data and figure supplement(s) for figure 5:

**Source data 1.** Summary table of genotypes, statistical testing, and graph data for *Figure 5*.

**Figure supplement 1.** Nociceptive rolling events induced by optogenetic activation.

**Figure supplement 1—source data 1.** Summary table of genotypes and graph data for *Figure 5—figure supplement 1*.

---

(*Figure 5A*, *Figure 5—figure supplement 1A and B*). In contrast, activation of SELK neurons induced robust rolling behavior in all tested larvae.

Based on this observation and *bero* knockdown results, we hypothesize that acute evoked activity of ABLK neurons may enhance nociceptive escape behavior. To test this, we silenced ABLK neurons by expressing the inward rectifying potassium channel Kir2.1, which mildly enhanced rolling escape behavior (*Figure 5D*). This result may be attributed to the silencing of persistent activity in ABLK neurons, which may disinhibit acute evoked nociceptive activity. Alternatively, chronic silencing of ABLK neurons may lead to modulation of nociceptive circuitry during development. Thus, in order to manipulate the activity of the neurons transiently, we combined the heat stimulation with optogenetic manipulation. We optically silenced or activated ABLK neurons during the first second of noxious thermal stimulation, such that the heat stimulation and optogenetic manipulation of the neurons were performed almost simultaneously (*Figure 5C*; 'Materials and methods'). Inhibition of ABLK neurons using *Guillardia theta anion channelrhodopsin-1* (GtACR1; *Mohammad et al., 2017*) significantly impaired rolling escape behavior (*Figure 5E*), suggesting that acute evoked activity of ABLK neurons is essential for nociceptive escape behavior. Unexpectedly, however, concurrent activation of ABLK neurons also resulted in significantly decreased rolling escape behavior (*Figure 5F*). These seemingly inconsistent results suggest that the neuronal activity induced by optogenetic activation is not comparable with the naturally evoked nociceptive responses in ABLK neurons, and proper dynamics of acute activity in ABLK neurons is essential for the facilitation of the rolling escape behavior. The rolling response of animals expressing myr:GFP in ABLK neurons was comparable in both ATR (+) and ATR (−) groups under light illumination, indicating that ATR does not influence the behavior (*Figure 5G*).

## DH44 and octopamine are functional neurotransmitters of ABLK neurons

We finally explored the functional neurotransmitters from ABLK neurons that enhance nociceptive behavior. Previous studies have reported that ABLK neurons produce leucokinin (LK; *de Haro et al., 2010*) and corticotropin-releasing factor (CRF)-like diuretic hormone (diuretic hormone 44, DH44; *Zandawala et al., 2018*). Our immunohistochemistry tests additionally identified that ABLK neurons express tyrosine decarboxylase 2 (Tdc2), a key synthetic enzyme for tyramine (TA) and octopamine (OA; *Figure 6A*). In contrast, ABLK neurons are neither GABAergic, glutamatergic, nor cholinergic (*Figure 6A*).

To test whether these neurotransmitters are required for the facilitation of nociceptive escape behavior, we used the intersectional flip-out approach to specifically suppress the expression of *Lk*, *Dh44* or *Tyramine β hydroxylase* (*Tbh*; a key synthetic enzyme converting tyramine to octopamine) in ABLK neurons by RNAi and examined the rolling escape behavior of these larvae in HPA experiments. We found that suppression of *Dh44* or *Tbh* reduced the rolling probability, indicating functional roles

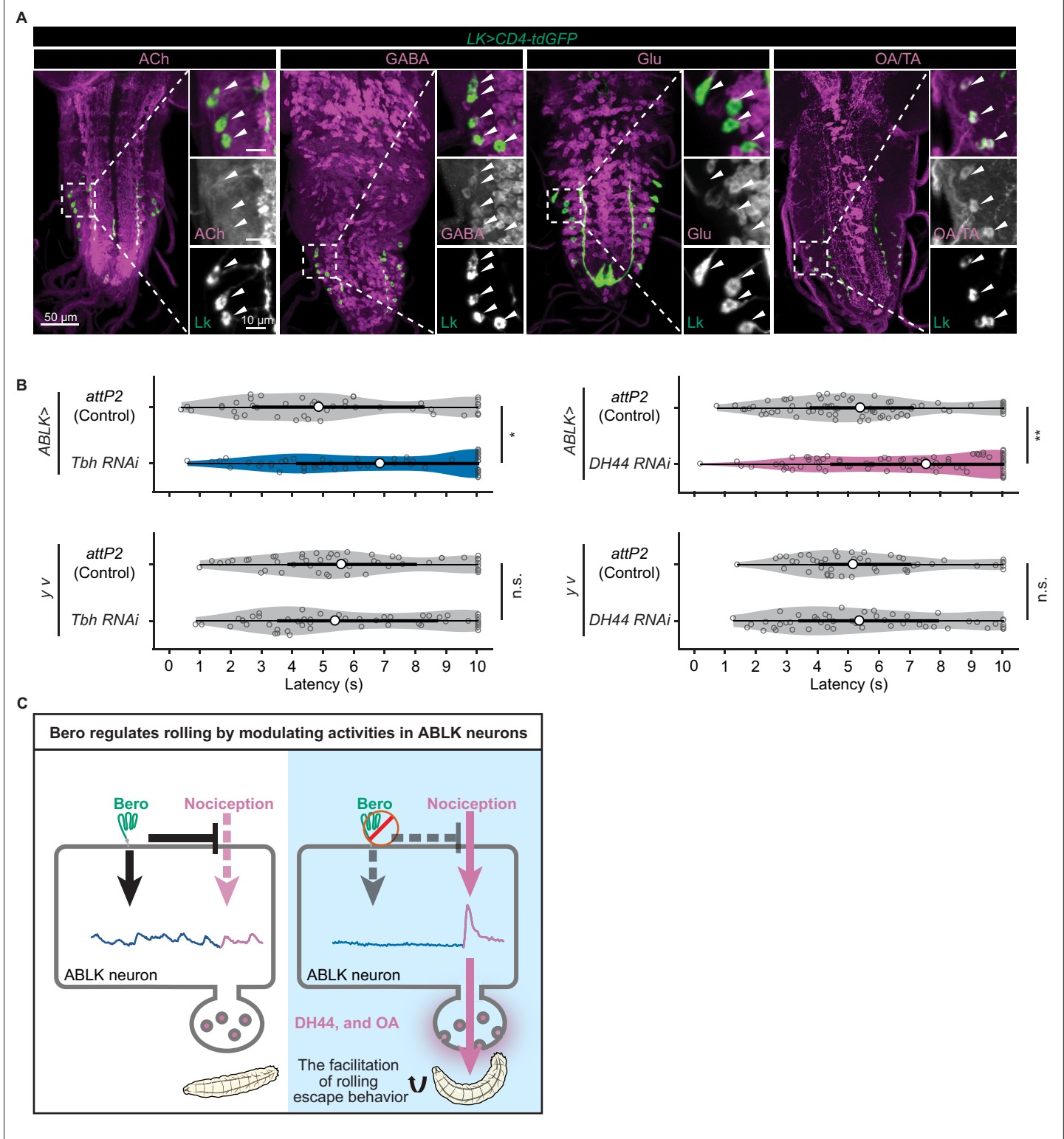

**Figure 6.** DH44 neuropeptides and octopamine are functional neurotransmitters of ABLK neurons. (**A**) Confocal image stacks showing LK neurons (*Lk-GAL4.TH*, *UAS-CD4-tdGFP*, green) and distinct types of neurons labeled by the corresponding neurotransmitter markers (cholinergic neurons, anti-ChAT; GABAergic neurons, anti-GABA; glutamatergic neurons, *VGlut-T2A-LexA>LexAop-jRCaMP1b*; tyraminergic and/or octopaminergic neurons, anti-Tdc2; magenta) in third-instar larvae. Enlarged panels show a magnified view of the boxed region. ABLK neurons are indicated by arrowheads. Scale bars: 50 μm; 10 μm for the magnified view. (**B**) Rolling latency of ABLK-specific *Tbh* knockdown larvae (*ABLK>Tbh* RNAi, n = 59), control larvae (*ABLK>attP2*, n = 45), and the corresponding effector-controls (*y v* background: *attP2* control, n = 52, *Tbh RNAi*, n = 51). Rolling latency of ABLK-specific

*Figure 6 continued on next page*

*Figure 6 continued*

*DH44* knockdown larvae (*ABLK>DH44* RNAi, n = 83), control larvae (*ABLK>attP2*, n = 79), and the corresponding effector-controls (*y v* background: *attP2* control, n = 48, *DH44* RNAi, n = 50). *p<0.05, **p<0.01; n.s., nonsignificant; Wilcoxon rank-sum test. (**C**) A model illustrating how Belly roll (Bero) regulates two distinct neuronal activities of a group of *bero*-expressing neurons, ABLK neurons, which initiate and facilitate the nociceptive escape behavior in *Drosophila melanogaster* larvae.

The online version of this article includes the following source data and figure supplement(s) for figure 6:

**Source data 1.** Summary table of genotypes, statistical testing, and graph data for *Figure 6*.

**Figure supplement 1.** Rolling latency of ABLK-specific *Lk*, *Rdl*, or *5-HT1B* knockdown larvae.

**Figure supplement 1—source data 1.** Summary table of genotypes, statistical testing, and graph data for *Figure 6—figure supplement 1*.

**Figure supplement 2.** A hypothetical model.

of DH44 and octopamine in the facilitation of nociceptive escape behavior (*Figure 6B*). However, in the case of *Lk* suppression, despite the reduction in rolling probability, the corresponding effector-control also showed significantly reduced rolling response, suggesting that the observed phenotype may be attributed to UAS transgene leaky expression (*Figure 6—figure supplement 1A*).

A previous study showed that knockdown of the ionotropic GABA$_A$ receptor Resistance to dieldrin (Rdl) in all LK neurons increased the probability of escape behaviors only at low intensity of nociceptor stimulation (*Hu et al., 2020*). Consistently, our result showed that ABLK-neuron-specific knockdown of *Rdl* enhanced rolling escape behavior upon lower thermal stimulations (temperature of the heat probe: 44°C; *Figure 6—figure supplement 1B*). Another previous study suggested that the 5-HT1B receptor regulates larval turning behavior by changing the periodic activity level of ABLK neurons (*Okusawa et al., 2014*). We examined the effects of ABLK-neuron-specific knockdown of *5-HT1B* on rolling escape behavior in HPA experiments and found that the suppression of *5-HT1B* did not show any significant effects (*Figure 6—figure supplement 1C*).

## Discussion
### Natural variation in escape behaviors

In this study, we revealed that *D. melanogaster* larvae display an extensive natural variation in nociceptive escape behaviors. The behavioral variations in a *Drosophila* population may provide it with adaptive advantages in ever-changing environments, especially when coexisting with geographically and seasonally variable parasitoid wasp species (*Bergland et al., 2014*; *Fleury et al., 2004*). In an environment with a high abundance of wasps, it might be adaptive for the larvae to be sensitized to noxious stimuli, whereas in an environment with fewer wasps, a high nociceptive sensitivity might provide only limited protection or even interfere with efficient execution of other essential behaviors such as foraging.

We identified nociception-associated genetic variants through an unbiased GWA analysis (*Figure 1—source data 1*), which has been employed to discover causative genes corresponding to a given phenotype in a straightforward manner (*Long et al., 2014*). A strongly associated SNP in the *bero* locus is located in an intronic region, suggesting that it affects gene expression. Indeed, we discovered a significant correlation between the SNP and the expression level of *bero*: animals with a homozygous cytosine allele have lower expression levels of *bero* and are more sensitive to noxious heat stimulations compared to those with a homozygous thymine allele. Consistently, pan-neuronal *bero* knockdown, *bero* heterozygous mutant (*bero*[KO]/*bero*[+]), and homozygous mutant (*bero*[KO]/*bero*[KO]) animals showed enhanced nociceptive rolling responses, implying that the reduction of the *bero* expression level has a profound impact on escape behavior. Notably, most homozygous mutant animals with the Canton-S background did not survive to adulthood, suggesting that *bero* plays an essential role in animal development as well as nociception.

In the validation experiments testing 17 candidate genes, we found that only one pan-neuronal RNAi knockdown produced a significant effect on nociceptive rolling escape behavior (*Figure 2A*). One interpretation of the low yield is that most of the variants identified by GWA analysis were false positives since the p-values associating these variants exceeded the Bonferroni-corrected 10% significance threshold, $6.95 \times 10^{-8}$; the alternative is that some polymorphisms in the DGRP strains are gain-of-function mutations. A third possibility is that some candidates are required only in non-neuronal

tissues for the behavior. We also noticed that the variants in genes essential for nociception, such as *Transient receptor potential cation channel A1* and *painless* hardly affected the behaviors. One reason is that these genes are essential for survival in natural environments and hence have no functional variants.

## How does *bero* control the activities of ABLK neurons?

Bero has been identified as a member of the Ly6/α-neurotoxin protein superfamily (*Hijazi et al., 2009*), whereas little is known about its biological functions (*Khan et al., 2017*). Notably, mammalian Ly6 proteins affect nicotinic acetylcholine receptors (nAChR) to modulate neuronal excitability (*Tsetlin, 2015*). A *Drosophila* Ly6-like protein, Quiver (Qvr)/Sleepless (Sss), plays dual roles in sleep regulation, not only suppressing neuronal excitability by enhancing the expression levels and open probability of Shaker potassium channels but also reducing cholinergic synaptic transmission by antagonizing nAChR (*Wu et al., 2014*; *Wu et al., 2010*). In our study, the knockdown of *bero* abolished persistent activity and increased nociceptive responses in ABLK neurons. At least two mechanisms could account for the role of *bero* in the two distinct neuronal activities. One interpretation could be that *bero* can regulate the two neuronal activities by modulating the functions of distinct transmembrane proteins separately. Alternatively, *bero* may target a single protein required for generating the persistent activities, which then constrains the evoked nociceptive responses in ABLK neurons and thereby downregulates the nociceptive rolling behavior (*Figure 6—figure supplement 2*). In the latter scenario, the *bero*-targeting protein may elevate intracellular calcium concentration, which interferes with the opening of voltage-gated cation channels, thereby suppressing nociceptive responses (*Budde et al., 2002*; *Oh et al., 2021*). Although our current model is based on preliminary observations, it may serve as a conceptual framework for future studies on the roles of *bero* and ABLK neurons in controlling nociceptive escape behaviors.

In both scenarios, a key issue is the identity of Bero-interacting membrane proteins in ABLK neurons. ABLK neurons express 5-hydroxytryptamine receptor 1B (5-HT1B; *Okusawa et al., 2014*), leucokinin receptor (*Okusawa et al., 2014*), insulin-like receptor (*Luo et al., 2013*), RYamide receptor (*Veenstra and Khammassi, 2017*), leucine-rich repeat-containing G protein-coupled receptor 4 (*Imambocus et al., 2022*), and Ecdysis-triggering hormone receptor (*Kim et al., 2006*). Among them, 5-HT1B has been reported to be required for the maintenance of persistent activity *Okusawa et al., 2014*; however, our preliminary results suggest that it is unlikely to be the effectors of Bero (*Figure 6—figure supplement 1C*). As such, the identity of the relevant Bero-interacting partners still needs to be established.

## ABLK neurons initiate the nociceptive escape behaviors

In this study, we identified ABLK neurons as a group of *bero*$^+$ neurons regulating nociceptive escape behaviors. Activation of all LK neurons was sufficient to induce robust rolling behavior (*Figure 5A*), which is consistent with previous studies (*Hu et al., 2020*; *Imambocus et al., 2022*). Furthermore, we succeeded in the specific activation of two distinct subgroups of LK neurons: ABLK or SELK neurons, both of which can be activated upon nociceptive stimulations (*Figure 4D*; *Hu et al., 2020*). A previous study hypothesized that LK neurons decode the decision-related activity, and the activation of LK neurons produces the nociceptive escape behavior (*Hu et al., 2020*). Here, we showed that the specific activation of ABLK neurons does not result in complete rolling, but rather robust bending behavior, which is considered to be an initial posture of rolling (*Burgos et al., 2018*; *Figure 5A*). Based on this observation, we hypothesize that the activation of ABLK neurons induces the initiation of rolling. Consistent with this idea, we found that *bero* knockdown in LK neurons resulted in elevated levels of nociceptive responses in ABLK neurons (*Figure 4D and F*) and concomitantly enhanced nociceptive rolling escape behavior (*Figure 3E*). Notably, activation of SELK neurons leads to the robust rolling behavior (*Figure 5A*); although this is not the major concern of this study, the pre- and postsynaptic circuitry of SELK neurons should be investigated as an important command circuitry for nociceptive rolling.

We also showed that either activation or inhibition of ABLK neurons during noxious thermal stimulation impairs rolling behavior (*Figure 5E and F*), suggesting that maintaining ABLK neuronal activity within an appropriate physiological range and/or dynamics is essential for promoting the behavior, supported by previous findings that a proper level of ABLK neuronal activity is crucial for controlling

noxious light-evoked turning behavior (*Okusawa et al., 2014*). More specifically, we propose a scenario that is based on the Ca$^{2+}$-dependent inhibition of transient excitation in ABLK neurons, as discussed above. In this scenario, transient activation of CsChrimson results in membrane depolarization and a large influx of Ca$^{2+}$ ions into the cytoplasm of ABLK neurons. This surge of Ca$^{2+}$ ions can suppress voltage-gated cation channels, thereby inhibiting subsequent nociceptive responses for a substantial period of time (*Budde et al., 2002*; *Oh et al., 2021*). Conversely, transient activation of GtACR1 suppresses nociceptive responses in the neurons, resulting in the nociceptive insensitivity. Another scenario proposes that activation of ABLK neurons may not only elicit immediate bending behavior but also exert a delayed suppressive effect on subsequent nociceptive responses to thermal stimulation through modulation of downstream circuits.

Previous research has showed that Kir2.1-mediated silencing of LK neurons increased the noxious touch response (*Imambocus et al., 2022*). Consistent with these findings, our results showed that Kir2.1-mediated silencing of ABLK neurons mildly enhanced rolling escape behavior in the HPA experiments (*Figure 5D*), which differs from effect of GtACR1-mediated inhibition (*Figure 5E*). We propose two possible scenarios. In the first scenario, Kir2.1-mediated hyperpolarization attenuates the persistent activities, thereby facilitating nociceptive responses. An alternative scenario suggests that continuous silencing may modulate the nociceptive circuitry during development, leading to increased nociceptive sensitivity. In contrast, GtACR1-mediated transient inhibition acutely interrupts nociceptive responses without affecting the network excitability.

## ABLK neurons as a potential neuromodulatory hub for nociceptive escape behaviors

A previous study reported that ABLK neurons showed evoked neuronal responses upon UV/blue light stimulation, and the responses were required for light-avoidance behavior (*Imambocus et al., 2022*). Of note, the roles of ABLK neurons in promoting noxious light-avoidance behavior and in initiating nociceptive rolling escape behavior do not necessarily conflict with each other; indeed, combined with nociceptor activation, the blue light stimulation can enhance nociceptive rolling escape behavior (*Wietek et al., 2017*). These findings imply that ABLK neurons receive converging inputs and affect differential outputs when animals encounter different modalities of noxious stimuli. Upon UV/blue light stimulation, for example, the evoked responses of ABLK neurons are employed for light-avoidance behavior; in contrast, upon noxious heat stimulation, the transient activities with appropriate dynamics facilitate the rolling. It will be interesting to test whether *bero* knockdown in ABLK neurons increases the UV light-induced activities in the neurons and enhances light-avoidance behavior.

ABLK neurons have known functions in the regulation of water balance and food intake in adults (*Liu et al., 2015*; *Zandawala et al., 2018*), suggesting that the persistent activities of the neurons may reflect stresses from desiccation and starvation. Therefore, if persistent activities suppress evoked nociceptive response in the neurons, *bero* would help integrate stresses with nociceptive inputs in the neurons. In other words, *bero* expressed in ABLK neurons could modulate nociceptive escape behavior based on the extent of stress in the animal.

Our study revealed that DH44 and octopamine released from ABLK neurons are required for the facilitation of nociceptive rolling behavior. DH44 is an ortholog of mammalian CRF. In mice, CRF$^+$ neurons in the central amygdala mediate conditioned flight (jump escape behavior; *Fadok et al., 2017*), where CRF release in the central amygdala increases mechanosensitivity and nocifensive reflexes (*Bourbia et al., 2010*; *Ji et al., 2013*; *Johnson et al., 2015*). Moreover, CRF mediates stress-induced thermal hyperalgesia in rats (*Itoga et al., 2016*), suggesting that the role of DH44 in facilitating nociceptive response might be evolutionarily conserved between insects and mammals.

In summary, our analyses revealed that *bero* modulates two distinct neuronal activities of a particular subset of *bero*-expressing neurons, ABLK neurons, which initiate and facilitate the nociceptive escape behavior in *D. melanogaster* larvae (*Figure 6C*). Our findings thus raise the possibility that *bero* has an essential role in monitoring stresses, which may modulate the nociceptive escape behavior (*Figure 6—figure supplement 2*).

## Materials and methods
### Resource availability
#### Lead contact
Further information and requests for resources and reagents should be directed to and will be fulfilled by the Lead Contact, Tadao Usui (usui.tadao.3c@kyoto-u.ac.jp).

#### Materials availability
Lines generated and described in this study are available on request from the lead contact.

#### Data and code availability
- All data reported in this paper are available from source data tables for each figure.
- All original code and scripts are available at this address: https://github.com/TadaoUsui/0_Bero (copy archived at *Usui and Li, 2023*).

### Experimental model and subject details
#### Fly stocks
*D. melanogaster* larvae were reared at 25°C and 75–80% humidity with a 12 hr light/dark cycle on standard fly food. Transgenic lines were mainly maintained in either *white* ($w^{1118}$) or *yellow vermillion* ($y^1$, $v^1$) mutant backgrounds unless stated otherwise. No sex-specific effects were part of this study. For fly line details, see Key Resources Table. Lines were obtained from the Bloomington Drosophila Stock Center or the Vienna Drosophila Stock Center unless stated otherwise. For behavioral analysis, wandering third-instar larvae of both sexes were used in this study (120 hr ± 12 hr AEL unless stated otherwise). For calcium imaging, female wandering third-instar larvae were used in this study. Detailed genotypes of experimental and control animals are listed in source data table of each figure.

### Method details
#### Generation of plasmids and transgenic strains
##### 20XUAS-Bero$^{FLAG}$
Bero cDNA was obtained from the *Drosophila* Genetics Resource Center (DGRC) and amplified from clone FI02856 (DGRC Stock 1621396; https://dgrc.bio.indiana.edu//stock/1621396; RRID:DGRC_1621396) by PCR. FLAG-tagged Bero was generated by inserting the FLAG sequence after the signal peptide sequence at position 26 of the Bero cDNA using overlap-PCR. Primers containing the FLAG-tag sequence were used for amplification and cloning into the pJFRC7-20XUAS-IVS-mCD8::GFP vector via NotI and XhaI restriction sites (the primer sequences are listed in *Supplementary file 1*). Transgene was made by φC31-mediated genomic integration into the *attP2* landing site (WellGenetics Inc, Taipei, Taiwan). pJFRC7-20XUAS-IVS-mCD8::GFP was a gift from Gerald Rubin (Addgene plasmid # 26220; http://n2t.net/addgene:26220; RRID:Addgene_26220).

##### VALIUM20-*bero*$^{sh\#1}$ and VALIUM20-*bero*$^{sh\#2}$
The *bero* RNAi lines were generated as described (*Ni et al., 2008*). Briefly, the annealed oligo DNA hybrid containing one of the 21-nt sequences targeting the coding region of *bero* (sh#1: GCACCAAGGACGAGTGCAACG; sh#2: CCTCTATGCCGTTCGTTAAGC) was cloned into a pVALIUM20 vector (the entire oligo DNA sequences are listed in *Supplementary file 1*). Transgene cassette insertion was achieved by φC31-mediated genomic integration into the *attP2* landing site on the 3rd chromosome (WellGenetics Inc).

#### Generation of *bero* knockout strains
CRISPR-mediated mutagenesis was performed by WellGenetics Inc using modified methods of *Kondo and Ueda, 2013*. In brief, the upstream gRNA sequence CAGACTGATCATAACGGCCA[CGG] and downstream gRNA sequence CATCCTGCTCTTCTTCGGCG[TGG] were cloned into U6 promoter plasmids separately. A *3xP3-RFP* cassette containing two *loxP* sites and two homology arms was cloned into pUC57 Kan as the donor template for repair (the primers used are listed in *Supplementary file 1*). *CG9336/bero* targeting gRNAs and *hs-Cas9* were supplied in DNA plasmids, together

with the donor plasmid for microinjection into embryos of control strain $w^{1118}$. F1 flies carrying selection marker of *3xP3-RFP* were further validated by genomic PCR and sequencing (the primers used are listed in **Supplementary file 1**). This CRISPR procedure generated a 1629 bp deletion allele in the *CG9336/bero* gene, replacing the entire CDS of the gene with the *3xP3-RFP* cassette (**Figure 2— figure supplement 1E**). This *bero* KO strain was then outcrossed to *Canton-S* for 11 generations and established as an isogenic line using *3xP3-RFP* as a selection marker.

## Immunohistochemistry and confocal imaging

Wandering third-instar larvae were dissected in PBS and fixed with 4% formaldehyde in PBS for 20 min at room temperature. After fixation, the larvae were washed five times with PBST (PBS containing 0.3% Triton X-100) and blocked in PBST containing 2% BSA (filtrated with 0.22 μm filter) for 30 min at room temperature. The samples were then incubated with primary antibodies at 4°C for 2 d. After five washes with PBST, the samples were incubated with the corresponding secondary antibodies for 1 hr at room temperature. After further washes, the samples were mounted with ProLong Glass Antifade Mountant (Thermo Fisher, Carlsbad, CA). Images were taken with a Nikon C1Si confocal microscope and processed with Fiji (ImageJ, NIH, Bethesda, MD).

The following primary antibodies were used: chicken polyclonal anti-GFP (1:1000; ab13970, Abcam); rat monoclonal anti-*D*N-cadherin (1:100; DN-Ex #8, Developmental Studies Hybridoma Bank [DSHB], Iowa City, IA); rabbit polyclonal anti-Lk (1:100; **Ohashi and Sakai, 2018**); mouse monoclonal anti-FLAG (M2) (1:500; F-3165, Sigma-Aldrich, Burlington, MA); rat monoclonal anti-DYKDDDDK Epitope Tag (L5) (1:500; NBP1-06712, Novus Biologicals, Littleton, CO); mouse monoclonal anti-ChAT (1:50; ChAT4B1, DSHB); rabbit polyclonal anti-GABA (1:100; A2052, Sigma-Aldrich); rabbit polyclonal anti-Tdc2 (1:1000; ab128225, Abcam, Cambridge, UK). Secondary antibodies were the following: Alexa Fluor 488-conjugated AffiniPure donkey polyclonal anti-chicken IgY (IgG) (H+L) (1:500; 703-545-155, Jackson ImmunoResearch Labs, West Grove, PA); Alexa Fluor 405-conjugated goat polyclonal anti-rat IgG (H+L) (1:500; ab175673, Abcam); Alexa Fluor 405-conjugated goat polyclonal anti-rabbit IgG (H+L) (1:500; A31556, Thermo Fisher Scientific); Alexa Fluor 546-conjugated goat polyclonal anti-mouse IgG (H+L) (1:500; A11030, Molecular Probes, Eugene, OR); Alexa Fluor 546-conjugated goat polyclonal anti-rat IgG (H+L) (1:500; A-11081, Molecular Probes); Alexa Fluor 555-conjugated goat polyclonal anti-mouse IgG (H+L) (1:500; A-21424, Thermo Fisher Scientific); Alexa Fluor 546-conjugated goat polyclonal anti-rabbit IgG (H+L) (1:500; A-11035, Thermo Fisher Scientific).

## Validation of *bero RNAi*<sup>shRNA#2</sup>

We validated the expression level of Bero (Bero-YFP, anti-GFP) in ABLK neurons in control and pan-neuronal *bero* knockdown larvae (*nSyb>bero RNAi<sup>shRNA#2</sup>*). *bero*-expressing neurons were labeled by *bero*-YFP (CPTI-001654) and immunohistochemistry (chicken anti-GFP, 1:1000; Alexa Fluor 488 anti-chicken, 1:500). The Immunohistochemistry was performed as described above. *z* stacks were obtained using a Nikon C1Si confocal microscope and processed with Fiji (ImageJ; NIH). Optical setting such as laser power and detector gain were identical for all samples. The cell body in individual ABLK neurons was manually selected as a region of interest (ROI). The normalized fluorescence was defined as $F_{\text{Bero-YFP}}/F_{\text{background}}$, where $F_{\text{Bero-YFP}}$ and $F_{\text{background}}$ are defined as the mean fluorescence intensity of the ROI and background region, respectively.

## Genome-wide association analysis

Rolling phenotypes with four distinct statistical metrics (**Figure 1—source data 1**) covering the 38 representative DGRP lines were submitted to the DGRP2 web-based analysis pipeline (http://dgrp2. gnets.ncsu.edu; **Huang et al., 2014**; **Mackay et al., 2012**). Utilizing a linear mixed model, the analysis first adjusts the phenotype for the effects of five major chromosomal inversions (In(2L)t, In(2R)NS, In(3R)P, In(3R)K, and In(3R)Mo) and *Wolbachia* infection (**Mackay et al., 2012**). Then the analysis pipeline performed association tests for individual genetic variants with minor allele frequencies of 0.05 or greater, running a linear mixed model using the FaST-LMM algorithm (**Lippert et al., 2011**). We defined GWA significantly associated genetic variants (**Figure 1—source data 2**) by a p-value of 1.0 × 10<sup>−5</sup>, which is a nominal threshold commonly used in previous DGRP studies (**Mackay and Huang, 2018**). The data were visualized with a custom program written in MATLAB (The MathWorks, Inc,

Natick, MA). All genes within 1 kb upstream or downstream of significant associated genetic variants were taken to be associated genes (*Figure 1—source data 2*).

## Gene expression analysis

To quantitate gene expression of *bero* in the central nervous system (CNS) across distinct strains, we examined expression levels in the pooled CNS from two wandering third-instar larvae for the following strains: *Canton-S* (control); four DGRP lines (RAL_208, RAL_315, RAL_391, RAL_705); and the *bero* KO strain which had been outcrossed to *Canton-S* for 11 generations. Total RNA from the respective CNS was extracted using Sepasol-RNA I (Nacalai tesque, Kyoto, Japan). Extracted RNA samples were further purified using RNeasy Mini Kit (QIAGEN N.V., Venlo, Netherlands). cDNA was synthesized from purified total RNA using ReverTra Ace qPCR RT Master Mix with gDNA Remover (Toyobo Co., Ltd., Osaka, Japan) following the manufacturer's instructions. RT-PCR was performed on Mastercycler gradient thermocycler (Eppendorf, Hamburg, Germany) using KOD-Plus-Neo (Toyobo). Expression levels of the target mRNA were normalized against those of $\alpha Tub84B$ in the same samples. The primers used are listed in *Supplementary file 1*. The original files of the full raw unedited gels and figures with the uncropped gels are available in *Figure 2—figure supplement 2—source data 2*.

## Heat Probe Assay (thermal nociception behavioral assay)

The thermal nociception behavioral assays were performed as previously described (*Onodera et al., 2017*; *Tracey et al., 2003*) with slight modifications. Animals were raised at 25°C in an incubator with 12 hr light/dark cycles on standard fly food. Humidity was manually controlled (75–80%). Wandering third-instar larvae of both sexes were picked up from the vial, rinsed twice with deionized water, and transferred to a 140 × 100 mm Petri dish with a bed of fresh 2% agarose gel. Larvae were touched laterally in abdominal segments (A4–A6) with a noxious heat probe (a soldering iron) at 46°C, unless otherwise stated.

The behavioral responses were recorded and later analyzed. The response latency was defined as the time elapsed between the delivery of the heat stimulation and the animals' initiation of the first 360° rotation. The response latency was computed as 10.05 s if the latency was more than 10 s. Sample sizes were estimated based on previous publications in the field (*Jovanic et al., 2016*; *Ohyama et al., 2015*). To reduce possible variation, each genotype was tested multiple times on different days, and the behavioral responses were analyzed in a blinded fashion. The data were visualized with a custom program written in MATLAB (The MathWorks, Inc).

## Optogenetic behavioral assays

Animals for optogenetic manipulation experiments were grown in the dark at 25°C for 5 d on fly food containing 0.5 mM all *trans*-retinal (R2500; Sigma-Aldrich) unless otherwise stated, and humidity was manually controlled (75–80%).

The optogenetic system consists of a dark chamber with a circuit that controls two arrays of LEDs placed in such a way as to uniformly cover the entire arena. To conduct the experiments, individual wandering third-instar larvae were placed on a 100 × 100 mm matte coated copper plate covered with 10 mL of deionized water. The larva was covered with the water film to be able to crawl on the plate. A 30-s-long pulse of 590 nm orange light (29.3 µW/mm²) was delivered (Amber LUXEON Rebel LED; Quadica Developments Inc, Alberta, Canada). The intensity at the water surface was measured using a HIOKI optical sensor (HIOKI E.E. CORPORATION, Nagano, Japan). Two infrared LED lights (LDR2-90IR2-850; CCS Inc, Kyoto, Japan) surrounding the plate allowed recording of the larval response in darkness. A custom software program written in LabVIEW (National Instruments, Austin, TX) and a multifunction DAQ device (NI USB-6210; National Instruments) were used to control the delivery of LED light pulses and record behavioral responses using a GE60 CCD imager (Library Co., Ltd, Tokyo, Japan). Videos were then examined manually and further analyzed automatically using FIMTrack (Version 2.1; https://github.com/kostasl/FIMTrack; *Risse et al., 2017*; *Lagogiannis, 2016*). We then performed the decoding analysis using a support vector machine (SVM)-based custom program written in MATLAB (The MathWorks, Inc) to determine whether each larva displayed rolling behavior in a frame-by-frame manner. Rolling was defined as at least one complete 360° rotation along the body axis. Bending was defined as a C-shape-like twitching without rolling, typically observed as

incomplete rolling behavior. Both manually and automatically examined data were visualized with a custom program written in MATLAB (The MathWorks, Inc).

To combine the optogenetic manipulation with the Heat Probe Assay, individual wandering third-instar larvae were placed in a 140 × 100 mm Petri dish as described. The heat stimulation and light illumination were delivered almost simultaneously: a 1-s-long light pulse of 590 nm orange LED (~30 µW/mm²; M590L3; ThorLabs, Newton, NJ) or 565 nm lime LED (~80 µW/mm²; M565L3; ThorLabs) was delivered using a TTL-controlled LED driver (LEDD1B; ThorLabs), respectively, for optogenetic activation or inhibition. The onset of the TTL pulse for the LED driver was synchronized semi-automatically with the heat stimulation using a microcontroller board (Arduino Nano Every; Arduino.cc, Lugano, Switzerland) with a custom-written program (Arduino IDE 2.0; Arduino). The behavioral responses were recorded and then analyzed offline as described.

## Locomotion assay

Animals for locomotion analysis were reared in the dark at 25°C for 4 d on fly food, and humidity was manually controlled (75–80%). For experiments, individual wandering larvae were placed on a 100 × 100 mm matte-coated aluminum plate covered with a bed of 2% agarose gel. Larval locomotion was recorded using a CMOS imager (EG130-B; Shodensha, Osaka, Japan) at a rate of 30 frames per second for 1 min. Larvae that exceeded the imaging area were excluded from the analysis. The loco-motor characteristics were automatically analyzed using FIMTrack. The total number of 'left-bended' and 'right-bended' frames was used to calculate the number of frames for head casting. Average velocity was calculated over 15-frame moving windows.

## Bioinformatic analysis of Bero

SignalP-5.0 (https://services.healthtech.dtu.dk/service.php?SignalP-5.0) identified the amino terminal signal peptide (*Almagro Armenteros et al., 2019*). NetNGlyc-1.0 (https://services.healthtech.dtu.dk/service.php?NetNGlyc-1.0) identified two possible glycosylation sites (*Gupta and Brunak, 2001*). Big-PI Predictor (https://mendel.imp.ac.at/gpi/gpi_server.html) predicted the GPI-anchor site (*Eisenhaber et al., 1999*). The putative transmembrane region was predicted using TMHMM-2.0 (https://services.healthtech.dtu.dk/services/TMHMM-2.0/) (*Krogh et al., 2001*). A three-dimensional model of Bero, along with putative disulfide bridges, was generated using AlphaFold2 and visualized with PyMOL (*Jumper et al., 2021*).

## Calcium imaging in larvae

Third wandering larvae were dissected in calcium-free external saline solution (120 mM NaCl, 3 mM KCl, 4 mM MgCl₂, 10 mM NaHCO₃, 5 mM TES, 10 mM HEPES, 10 mM glucose, 10 mM sucrose, 10 mM trehalose, 1 mM sodium L-glutamate; osmolality was adjusted to 305 mOsm/kg with water, and the final pH was adjusted to 7.25 with NaOH; modified from *Xiang et al., 2010*). Buffer was exchanged with oxygenated external saline solution (1.5 mM CaCl₂ added to calcium-free external saline solution). The cell bodies were imaged at one frame per second.

470 nm LED blue light (M470L5; ThorLabs) was applied 250 s after the start of imaging and delivered using a TTL-controlled LED driver (LEDD1B; ThorLabs) for optogenetic activation or inhibition. The onset of the TTL pulse for the LED driver was operated using a microcontroller board (Arduino Uno R3; Arduino.cc) with a custom-written program (Arduino IDE 1.0; Arduino.cc). Blue light was applied at 0.37 mW/mm² for 2.5 s. jRCaMP1b was excited with a 561 nm diode laser (Sapphire 561 LP; Coherent). Images ware acquired at 512 × 512 pixels, with a 250 ms exposure time. The fluorescence signal was captured with images at 1 fps through 610/60 nm bandpass filters (Chroma Technology Corp., Bellows Falls, VT) using an inverted microscope (IX-71; Olympus, Tokyo, Japan) equipped with an EMCCD camera (iXon X3; Andor Technology Ltd., Belfast, UK) with a ×20 objective lens (UCPlanFLN ×20/0.45; Olympus). The data were analyzed with a custom program written in MATLAB (The MathWorks, Inc). The cell body in individual neurons was manually selected as an ROI from the confocal time series.

For the analysis of the persistent activity, the fluorescence change was defined as

$$\frac{\Delta F^{\text{Persistent}}}{F_0} = \frac{F(t) - (B(t) - B_0(t)) - F_0(t)}{F_0(t)} \times 100$$

where $F(t)$ and $B(t)$ are the fluorescence intensity of the ROI and background region at time point $t$, respectively; $F_0(t)$ and $B_0(t)$ are defined as the median of the five minimal values of fluorescence intensity of the ROI and background region, respectively, from $t = -10$ s to $t = 10$ s. The sum of $\Delta F^{\text{Persistent}}/F_0(t)$ values was used as the area under the curve (AUC).

For the analysis of the evoked activities, the fluorescence change was defined as

$$\frac{\Delta F}{F_0} = \frac{F(t) - (B(t) - B_0) - F_0}{F_0} \times 100$$

where $F(t)$ and $B(t)$ are the fluorescence intensity of the ROI and background region at time point $t$, respectively; $F_0$ and $B_0$ are defined as the median of the five minimal values of fluorescence intensity of the ROI and background region, respectively, during a 50 s period just before blue light stimulation was initiated. $F_0$ was used as the baseline fluorescence. The maximum intensity of $\Delta F/F_0(t)$ from $t = 0$ s to $t = 5$ s was used as the $\Delta F_{\text{max}}/F_0$.

## Quantification and statistical analysis

### Statistics

Sample sizes were similar to those of previous studies. For comparisons of two groups, an unpaired Welch's $t$-test was used, or a nonparametric Wilcoxon rank-sum test, as noted in figure legends. For multiple comparison, a Kruskal–Wallis test followed by Dunn's multiple-comparison test was used. For comparisons of optogenetic activation-induced behaviors, a Chi-squared test was used. Differences were considered significant for $p<0.05$ (*$p<0.05$, **$p<0.01$, ***$p<0.001$, ****$p<0.0001$). Statistical testing was performed using MATLAB (The MathWorks, Inc) or R (version 3.6.1, R Development Core Team). Exact statistical tests and p-values for all quantitative data comparisons are listed in the source data tables.

## Acknowledgements

We thank M Futamata, S Oki, and H Imai for excellent technical assistance; members of the Uemura lab for experimental advice and discussions; J Hejna for feedback on the manuscript; T Sakai (Tokyo Metropolitan Univ.) for the anti-Lk primary antibody; J Simpson (UCSB) for tsh-LexA and Scr-LexA strains; A Claridge-Chang (Duke-NUS Medical School) and K Emoto (U Tokyo) for UAS-GtACR1 strain; *Drosophila* Stocks of Ehime University, Kyoto *Drosophila* Stock Center, and Bloomington *Drosophila* Stock Center for providing fly stocks. We also thank FlyBase and *Drosophila* Genomics Resource Center. This study was supported by KAKENHI from the Japan Society for the Promotion of Science (JSPS) grants 21K06264 to TUs and 15H02400 to TUe, by Toray Science and Technology Grant from Toray Science Foundation to TUs and by a JSPS Research Fellowship for Young Scientists to YT.

## Additional information

### Funding

| Funder | Grant reference number | Author |
| --- | --- | --- |
| Japan Society for the Promotion of Science | 21K06264 | Tadao Usui |
| Toray Science Foundation | Toray Science and Technology Grant | Tadao Usui |
| Japan Society for the Promotion of Science | 15H02400 | Tadashi Uemura |
| Japan Society for the Promotion of Science | JSPS Research Fellowship for Young Scientist | Yuma Tsukasa |

The funders had no role in study design, data collection and interpretation, or the decision to submit the work for publication.

## Author contributions
Kai Li, Conceptualization, Resources, Data curation, Software, Formal analysis, Validation, Investigation, Visualization, Methodology, Writing - original draft, Writing - review and editing; Yuma Tsukasa, Conceptualization, Resources, Data curation, Software, Formal analysis, Funding acquisition, Validation, Investigation, Visualization, Methodology, Writing - review and editing; Misato Kurio, Kaho Maeta, Methodology; Akimitsu Tsumadori, Software, Methodology, Writing - review and editing; Shumpei Baba, Akira Murakami, Software, Methodology; Risa Nishimura, Resources, Data curation, Formal analysis, Investigation, Methodology; Koun Onodera, Data curation, Software, Investigation, Methodology, Writing - review and editing; Takako Morimoto, Methodology, Writing - review and editing; Tadashi Uemura, Funding acquisition, Writing - review and editing; Tadao Usui, Conceptualization, Resources, Software, Supervision, Funding acquisition, Validation, Investigation, Methodology, Writing - original draft, Project administration, Writing - review and editing

## Author ORCIDs
Yuma Tsukasa ⓘ http://orcid.org/0009-0007-3690-4000
Koun Onodera ⓘ http://orcid.org/0000-0002-4203-9865
Tadao Usui ⓘ http://orcid.org/0000-0002-0507-1495

## Decision letter and Author response
Decision letter https://doi.org/10.7554/eLife.83856.sa1
Author response https://doi.org/10.7554/eLife.83856.sa2

## Additional files

### Supplementary files
- Supplementary file 1. Oligonucleotide sequences.
- Supplementary file 2. Basic sequence analysis of 2L_20859945_SNP neighboring sequence.
- MDAR checklist

### Data availability
All of the source data are submitted as part of our manuscript.

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

# Appendix 1

## Appendix 1—key resources table

| Reagent type (species) or resource | Designation | Source or reference | Identifiers | Additional information |
|---|---|---|---|---|
| Gene (*Drosophila melanogaster*) | CG9336 | GenBank | FLYB:FBgn0032897 | |
| Strain, strain background (*Escherichia coli*, DH5α) | E coli | ATCC | ATCC:PTA-1798 | DH5α is an *E. coli* strain used for general cloning applications |
| Strain, strain background (*D. melanogaster*) | w[1118] | Bloomington *Drosophila* Stock Center | BDSC:3605; FLYB:FBst0003605; RRID:BDSC_3605 | |
| Strain, strain background (*D. melanogaster*) | Canton-S | *Drosophila* Stocks of Ehime University | DSEU:E-10002 | |
| Strain, strain background (*D. melanogaster*) | 38 representative DGRP inbred strains | Bloomington *Drosophila* Stock Center | N/A | See *Figure 1—source data 1* in this paper |
| Genetic reagent (*D. melanogaster*) | nSyb; nSyb-Gal4 | Bloomington *Drosophila* Stock Center | BDSC:51941; FLYB:FBtp0087725; RRID:BDSC_51941 | FlyBase symbol: P{nSyb-GAL4.P} |
| Genetic reagent (*D. melanogaster*) | eys; eys RNAi | Bloomington *Drosophila* Stock Center | BDSC:33764; FLYB:FBti0141009; RRID:BDSC_33764 | FlyBase symbol: P{TRiP.JF02463}attP2 |
| Genetic reagent (*D. melanogaster*) | eys; eys RNAi | Bloomington *Drosophila* Stock Center | BDSC:33766; FLYB:FBti0141011; RRID:BDSC_33766 | FlyBase symbol: P{TRiP.JF02708}attP2 |
| Genetic reagent (*D. melanogaster*) | G9336; bero RNAi[JF] | Bloomington *Drosophila* Stock Center | BDSC:31988; FLYB:FBti0130397; RRID:BDSC_31988 | FlyBase symbol: P{TRiP.JF03422}attP2 |
| Genetic reagent (*D. melanogaster*) | luna; luna RNAi | Bloomington *Drosophila* Stock Center | BDSC:27084; FLYB:FBti0115451; RRID:BDSC_27084 | FlyBase symbol: P{TRiP.JF02430}attP2 |
| Genetic reagent (*D. melanogaster*) | Prosap; Prosap RNAi | Bloomington *Drosophila* Stock Center | BDSC:40929; FLYB:FBti0149838; RRID:BDSC_40929 | FlyBase symbol: P{TRiP.HMS02177}attP2 |
| Genetic reagent (*D. melanogaster*) | CG14669; CG14669 RNAi | Bloomington *Drosophila* Stock Center | BDSC:36750; FLYB:FBti0146797; RRID:BDSC_ 36750 | FlyBase symbol: P{TRiP.HMS03010}attP2 |
| Genetic reagent (*D. melanogaster*) | gukh; gukh RNAi | Bloomington *Drosophila* Stock Center | BDSC:55858; FLYB:FBti0163218; RRID:BDSC_ 55858 | FlyBase symbol: P{TRiP.HMC03681}attP2 |
| Genetic reagent (*D. melanogaster*) | Antp; Antp RNAi | Bloomington *Drosophila* Stock Center | BDSC:64926; FLYB:FBti0184012; RRID:BDSC_ 64926 | FlyBase symbol: P{TRiP.HMC05799}attP2 |
| Genetic reagent (*D. melanogaster*) | Erk7; Erk7 RNAi | Bloomington *Drosophila* Stock Center | BDSC:56939; FLYB:FBti0163468; RRID:BDSC_ 56939 | FlyBase symbol: P{TRiP.HMC04378}attP2 |
| Genetic reagent (*D. melanogaster*) | 5-HT1A; 5-HT1A RNAi | Bloomington *Drosophila* Stock Center | BDSC:25834; FLYB:FBti0114585; RRID:BDSC_ 25834 | FlyBase symbol: P{TRiP.JF01852}attP2 |
| Genetic reagent (*D. melanogaster*) | bc10; bc10 RNAi | Bloomington *Drosophila* Stock Center | BDSC:60486; FLYB:FBti0179272; RRID:BDSC_ 60486 | FlyBase symbol: P{TRiP.HMJ22879}attP40 |
| Genetic reagent (*D. melanogaster*) | CG31760; CG31760 RNAi | Bloomington *Drosophila* Stock Center | BDSC:51838; FLYB:FBti0157803; RRID:BDSC_ 51838 | FlyBase symbol: P{TRiP.HMC03410}attP2 |
| Genetic reagent (*D. melanogaster*) | CG4168; CG4168 RNAi | Bloomington *Drosophila* Stock Center | BDSC:28736; FLYB:FBti0127300; RRID:BDSC_ 28736 | FlyBase symbol: P{TRiP.JF03163}attP2 |

*Appendix 1 Continued on next page*

*Appendix 1 Continued*

| Reagent type (species) or resource | Designation | Source or reference | Identifiers | Additional information |
|---|---|---|---|---|
| Genetic reagent (*D. melanogaster*) | CG43897; CG43897 RNAi | Bloomington *Drosophila* Stock Center | BDSC:31560; FLYB:FBti0130595; RRID:BDSC_31560 | FlyBase symbol: P{TRiP. JF01132}attP2 |
| Genetic reagent (*D. melanogaster*) | bru3; bru3 RNAi | Bloomington *Drosophila* Stock Center | BDSC:43318; FLYB:FBti0151331 RRID:BDSC_43318 | FlyBase symbol: P{TRiP. HMS02702}attP40 |
| Genetic reagent (*D. melanogaster*) | Gyc88E; Gyc88E RNAi | Bloomington *Drosophila* Stock Center | BDSC:28608; FLYB:FBti0127063; RRID:BDSC_28608 | FlyBase symbol: P{TRiP. HM05096}attP2 |
| Genetic reagent (*D. melanogaster*) | olf413; olf413 RNAi | Bloomington *Drosophila* Stock Center | BDSC: 29547; FLYB:FBti0128663; RRID:BDSC_29547 | FlyBase symbol: P{TRiP. JF02439}attP2 |
| Genetic reagent (*D. melanogaster*) | bero RNAi[shRNA#1]; | This paper | N/A | See Materials and methods |
| Genetic reagent (*D. melanogaster*) | bero RNAi[shRNA#2]; | This paper | N/A | See Materials and methods |
| Genetic reagent (*D. melanogaster*) | nSyb; nSyb-GAL4 | Bloomington *Drosophila* Stock Center | BDSC:39171; FLYB:FBti0136953; RRID:BDSC_39171 | FlyBase symbol: P{GMR56C10-GAL4}attP2 |
| Genetic reagent (*D. melanogaster*) | bero-YFP; CG9336(CPTI001654) | Kyoto *Drosophila* Stock Center | DGRC:115180; FLYB:FBti0143870; RRID:DGGR_115180 | FlyBase symbol: PBac{681.P.FSVS-1} CG9336[CPTI001654] |
| Genetic reagent (*D. melanogaster*) | bero[KO]; bero knockout | This paper | N/A | See Materials and methods |
| Genetic reagent (*D. melanogaster*) | UAS-CD4-tdTomato; CD4-tdTomato; CD4-RFP | Bloomington *Drosophila* Stock Center | BDSC:35837; FLYB:FBti0143424; RRID:BDSC_35837 | FlyBase symbol: PBac{UAS-CD4-tdTom}VK00033 |
| Genetic reagent (*D. melanogaster*) | R72F11-GAL4; Basin-GAL4; Basin | Bloomington *Drosophila* Stock Center | BDSC:39786; FLYB:FBti0138028; RRID:BDSC_39786 | FlyBase symbol: P{GMR72F11-GAL4}attP2 |
| Genetic reagent (*D. melanogaster*) | Ilp2-GAL4.R; Ilp2 | Bloomington *Drosophila* Stock Center | BDSC:37516; FLYB:FBti0147109 RRID:BDSC_37516 | FlyBase symbol: P{Ilp2-GAL4.R}2 |
| Genetic reagent (*D. melanogaster*) | Eh.2.4-GAL4; Eh | Bloomington *Drosophila* Stock Center | BDSC:6301; FLYB:FBti0012534; RRID:BDSC_6301 | FlyBase symbol: P{GAL4-Eh.2.4}C21 |
| Genetic reagent (*D. melanogaster*) | dimm-GAL4; dimm | Bloomington *Drosophila* Stock Center | BDSC:25373; FLYB:FBti0004282; RRID:BDSC_25373 | FlyBase symbol: P{GawB} dimm[929] |
| Genetic reagent (*D. melanogaster*) | Lk-GAL4; Lk | Bloomington *Drosophila* Stock Center | BDSC:51993; FLYB:FBti0154847; RRID:BDSC_51993 | FlyBase symbol: P{Lk-GAL4. TH}2M |
| Genetic reagent (*D. melanogaster*) | ppk-GAL4; ppk | Bloomington *Drosophila* Stock Center | BDSC:32079; FLYB:FBti0131208; RRID:BDSC_32079 | FlyBase symbol: P{ppk-GAL4.G}3 |
| Genetic reagent (*D. melanogaster*) | R69F06-GAL4; Goro-GAL4 | Bloomington *Drosophila* Stock Center | BDSC:39497; FLYB:FBti0137775; RRID:BDSC_39497 | FlyBase symbol: P{GMR69F06-GAL4}attP2 |
| Genetic reagent (*D. melanogaster*) | tsh-LexA | DOI:10.1080/01677063.2016.1248761 | N/A | J. Simpson, UCSB, Santa Barbara, USA |
| Genetic reagent (*D. melanogaster*) | Scr-LexA | DOI:10.1080/01677063.2016.1248761 | N/A | J. Simpson, UCSB, Santa Barbara, USA |
| Genetic reagent (*D. melanogaster*) | tubP-FRT-GAL80-FRT | Bloomington *Drosophila* Stock Center | BDSC:38881; FLYB:FBti0147582; RRID:BDSC_38881 | FlyBase symbol: P{αTub84B(FRT.GAL80)}3 |
| Genetic reagent (*D. melanogaster*) | tubP-FRT-GAL80-FRT | Bloomington *Drosophila* Stock Center | BDSC:38880; FLYB:FBti0147581; RRID:BDSC_38880 | FlyBase symbol: P{αTub84B(FRT.GAL80)}2 |
| Genetic reagent (*D. melanogaster*) | LexAop-FLP.L | Bloomington *Drosophila* Stock Center | BDSC:55820; FLYB:FBti0160802; RRID:BDSC_55820 | FlyBase symbol: P{8XLexAop2-FLPL}attP40 |

*Appendix 1 Continued on next page*

*Appendix 1 Continued*

| Reagent type (species) or resource | Designation | Source or reference | Identifiers | Additional information |
|---|---|---|---|---|
| Genetic reagent (*D. melanogaster*) | LexAop-FLP.L | Bloomington *Drosophila* Stock Center | BDSC:55819; FLYB:FBti0160801; RRID:BDSC_55819 | FlyBase symbol: P{8XLexAop2-FLPL}attP2 |
| Genetic reagent (*D. melanogaster*) | UAS-Bero:FLAG; Bero:FLAG | This paper | N/A | See Materials and methods |
| Genetic reagent (*D. melanogaster*) | UAS-myr:GFP; myr:GFP | Bloomington *Drosophila* Stock Center | BDSC:32197; FLYB:FBti0131941; RRID:BDSC_32197 | FlyBase symbol: P{10XUAS-IVS-myr::GFP}attP2 |
| Genetic reagent (*D. melanogaster*) | UAS-CsChrimson; CsChrimson; Chrimson | Bloomington *Drosophila* Stock Center | BDSC:55135; FLYB:FBti0160803; RRID:BDSC_55135 | FlyBase symbol: P{20XUAS-IVS-CsChrimson.mVenus}attP40 |
| Genetic reagent (*D. melanogaster*) | UAS-CsChrimson; CsChrimson; Chrimson | Bloomington *Drosophila* Stock Center | BDSC:55136; FLYB:FBti0160804; RRID:BDSC_55136 | FlyBase symbol: P{20XUAS-IVS-CsChrimson.mVenus}attP2 |
| Genetic reagent (*D. melanogaster*) | UAS-GtACR1-EYFP | DOI: 10.1038/nmeth.4148; *Mohammad et al., 2017* | NA | A. Claridge-Chang, Duke-NUS Medical School, Singapore |
| Genetic reagent (*D. melanogaster*) | UAS-CD4-tdGFP | Bloomington *Drosophila* Stock Center | BDSC:35839; FLYB:FBti0143426 RRID:BDSC_35839 | FlyBase symbol: P{UAS-CD4-tdGFP}8M2 |
| Genetic reagent (*D. melanogaster*) | VGlut-LexA | Bloomington *Drosophila* Stock Center | BDSC:84442; FLYB:FBti0209982; RRID:BDSC_84442 | FlyBase symbol: TI{2A-lexA::GAD}VGlut$^{2A\text{-}lexA}$ |
| Genetic reagent (*D. melanogaster*) | jRCaMP1b; UAS- jRCaMP1b | Bloomington *Drosophila* Stock Center | BDSC:63793; FLYB:FBti0180189; RRID:BDSC_63793 | FlyBase symbol: PBac{20XUAS-IVS-NES-jRCaMP1b-p10}VK00005 |
| Genetic reagent (*D. melanogaster*) | jRCaMP1b; UAS- jRCaMP1b | Bloomington *Drosophila* Stock Center | BDSC:64428; FLYB:FBti0181971; RRID:BDSC_64428 | FlyBase symbol: P{13XLexAop2-IVS-NES-jRCaMP1b-p10}su(Hw)attP5 |
| Genetic reagent (*D. melanogaster*) | ChR2.T159C | Bloomington *Drosophila* Stock Center | BDSC:52259; FLYB:FBti0157028; RRID:BDSC_52259 | FlyBase symbol: PBac{10XQUAS-ChR2.T159C-HA}VK00018 |
| Genetic reagent (*D. melanogaster*) | Lk RNAi; UAS-Lk-RNAi | Bloomington *Drosophila* Stock Center | BDSC:25798; FLYB:FBti0114549; RRID:BDSC_25798 | FlyBase symbol: P{TRiP.JF01816}attP2 |
| Genetic reagent (*D. melanogaster*) | DH44 RNAi; UAS-Dh44-RNAi | Bloomington *Drosophila* Stock Center | BDSC:25804; FLYB:FBti0114555; RRID:BDSC_25804 | FlyBase symbol: P{TRiP.JF01822}attP2 |
| Genetic reagent (*D. melanogaster*) | Tbh RNAi; UAS-Tbh-RNAi | Bloomington *Drosophila* Stock Center | BDSC: 27667; FLYB:FBti0128848; RRID:BDSC_27667 | FlyBase symbol: P{TRiP.JF02746}attP2 |
| Genetic reagent (*D. melanogaster*) | TrpA1-QF | Bloomington *Drosophila* Stock Center | BDSC:36348; FLYB:FBti0145127; RRID:BDSC_36348 | FlyBase symbol: P{TrpA1-QF.P}attP40 |
| Genetic reagent (*D. melanogaster*) | attP2 | Bloomington *Drosophila* Stock Center | BDSC:8622; FLYB:FBti0040535; RRID:BDSC_8622 | FlyBase symbol: P{CaryP} attP2 |
| Genetic reagent (*D. melanogaster*) | attP40 | Bloomington *Drosophila* Stock Center | BDSC:36304; FLYB:FBti0114379; RRID:BDSC_36304 | FlyBase symbol: P{CaryP} Msp300$^{attP40}$ |
| Genetic reagent (*D. melanogaster*) | UAS-Kir2.1; Kir2.1 | Bloomington *Drosophila* Stock Center | BDSC:6595; FLYB:FBti0017552; RRID:BDSC_6595 | FlyBase symbol: P{UAS-Hsap\KCNJ2.EGFP}7 |
| Genetic reagent (*D. melanogaster*) | Rdl RNAi; UAS-Rdl-RNAi | Bloomington *Drosophila* Stock Center | BDSC:52903; FLYB:FBti0158020; RRID:BDSC_52903 | FlyBase symbol: P{TRiP.HMC03643}attP40 |
| Genetic reagent (*D. melanogaster*) | 5-HT1B RNAi; UAS-5-HT1B-RNAi | Bloomington *Drosophila* Stock Center | BDSC:25833; FLYB:FBti0114584; RRID:BDSC_25833 | FlyBase symbol: P{TRiP.JF01851}attP2 |
| Genetic reagent (*D. melanogaster*) | UAS-DenMark; DenMark | Bloomington *Drosophila* Stock Center | BDSC:33065; FLYB:FBti0132510; RRID:BDSC_33065 | FlyBase symbol: P{UAS-DenMark}3 |

*Appendix 1 Continued on next page*

Appendix 1 Continued

| Reagent type (species) or resource | Designation | Source or reference | Identifiers | Additional information |
|---|---|---|---|---|
| Genetic reagent (*D. melanogaster*) | UAS-syt:eGFP; syt:GFP | Bloomington *Drosophila* Stock Center | BDSC:33065; FLYB:FBti0026975; RRID:BDSC_33065 | FlyBase symbol: P{UAS-syt. eGFP}3 |
| Antibody | Anti-GFP (chicken polyclonal) | Abcam | Cat#: ab13970; RRID:AB_300798 | IF(1:1000) |
| Antibody | Anti-DN-cadherin (rat monoclonal) | Developmental Studies Hybridoma Bank | Cat#: DN-Ex #8; RRID: AB_528121 | IF(1:100) |
| Antibody | Anti-Lk (rabbit polyclonal) | DOI:10.1016/j.bbrc.2018.03.132; *Ohashi and Sakai, 2018* | N/A | IF(1:100) |
| Antibody | Anti-FLAG (mouse monoclonal, M2) | Sigma-Aldrich | Cat#: F3165; RRID:AB_259529 | IF(1:500) |
| Antibody | Anti-ChAT (mouse monoclonal) | Developmental Studies Hybridoma Bank | Cat#: chat4b1; RRID:AB_528122 | IF(1:50) |
| Antibody | Anti-GABA (rabbit polyclonal) | Sigma-Aldrich | Cat#: A2052; RRID: AB_477652 | IF(1:100) |
| Antibody | Anti-Tdc2 (rabbit polyclonal) | Abcam | Cat#: ab128225; RRID:AB_11142389 | IF(1:1000) |
| Antibody | Anti-DYKDDDDK Epitope Tag (rat monoclonal, L5) | Novus Biologicals | Cat#: NBP1-06712; RRID:AB_1625981 | IF(1:500) |
| Antibody | Alexa Fluor 488-conjugated AffiniPure anti-chicken IgY (IgG) (H+L) (donkey polyclonal) | Jackson ImmunoResearch Laboratories Inc | Cat#: 703-545-155; RRID:AB_2340375 | IF(1:500) |
| Antibody | Alexa Fluor 405-conjugated anti-rat IgG (H+L) (goat polyclonal) | Abcam | Cat#: ab175673; RRID:AB_2893021 | IF(1:500) |
| Antibody | Alexa Fluor 405-conjugated anti-rabbit IgG (H+L) (goat polyclonal) | Thermo Fisher Scientific | Cat# :A-31556; RRID:AB_221605 | IF(1:500) |
| Antibody | Alexa Fluor 546-conjugated anti-rat IgG (H+L) (goat polyclonal) | Molecular Probes | Cat#: A-11081; RRID:AB_141738 | IF(1:500) |
| Antibody | Alexa Fluor 546-conjugated anti-mouse IgG (H+L) (goat polyclonal) | Molecular Probes | Cat#: A-11030; RRID:AB_144695 | IF(1:500) |
| Antibody | Alexa Fluor 546-conjugated anti-rabbit IgG (H+L) (goat polyclonal) | Thermo Fisher Scientific | Cat#: A-11035; RRID:AB_2534093 | IF(1:500) |
| Antibody | Alexa Fluor 555-conjugated anti-mouse IgG (H+L) (goat polyclonal) | Thermo Fisher Scientific | Cat#: A-21424; RRID:AB_141780 | IF(1:500) |
| Recombinant DNA reagent | pJFRC7-mCD8::GFP (plasmid) | Addgene | RRID:Addgene_26220 | |
| Recombinant DNA reagent | pJFRC7-FLAG::Bero (plasmid) | This paper | N/A | FLAG::Bero version of pJFRC7-mCD8::GFP; see Materials and methods |
| Recombinant DNA reagent | pVALIUM20 (plasmid) | *Drosophila* Genomics Resource Center | RRID:DGRC_1467 | |
| Recombinant DNA reagent | pVALIUM20-bero [shRNA#1] (plasmid) | This paper | N/A | See Materials and methods |
| Recombinant DNA reagent | pVALIUM20-bero [shRNA#2] (plasmid) | This paper | N/A | See Materials and methods |
| Recombinant DNA reagent | FI02856 (cDNA) | *Drosophila* Genomics Resource Center | RRID:DGRC_1621396 | |
| Sequence-based reagent | Primers for bero and αTub84B | This paper | N/A | See *Supplementary file 1* |
| Sequence-based reagent | Oligo DNA sequence for bero shRNA#1 and shRNA#2 | This paper | N/A | See *Supplementary file 1* |
| Sequence-based reagent | gRNA sequence for bero knockout, see *Supplementary file 1* | This paper | N/A | See *Supplementary file 1* |
| Sequence-based reagent | Primers for homology arm (bero knockout) | This paper | N/A | See *Supplementary file 1* |

Appendix 1 Continued on next page

*Appendix 1 Continued*

| Reagent type (species) or resource | Designation | Source or reference | Identifiers | Additional information |
|---|---|---|---|---|
| Sequence-based reagent | Primers for validation PCR (bero knockout) | This paper | N/A | See *Supplementary file 1* |
| Commercial assay or kit | Sepasol-RNA I Super G | Nacalai tesque | Cat#: 09379-26 | |
| Commercial assay or kit | RNeasy Mini Kit | QIAGEN | Cat#: 74104 | |
| Commercial assay or kit | ReverTra Ace qPCR RT Master Mix with gDNA Remover | Toyobo | Cat#: FSQ-301 | |
| Commercial assay or kit | KOD-Plus-Neo | Toyobo | Cat#: KOD-401 | |
| Chemical compound, drug | all-trans retinal | Sigma-Aldrich | Cat#: R2500 | |
| Software, algorithm | Arduino IDE 1.8.13 or 2.0.0 | arduino.cc, https://www.arduino.cc/en/software/ReleaseNotes | N/A | |
| Software, algorithm | Fiji | NIH, Bethesda | RRID:SCR_002285 | |
| Software, algorithm | FimTrack | DOI:10.3791/52207; *Risse et al., 2017* | N/A | Version 2.1; https://github.com/kostasl/FIMTrack |
| Software, algorithm | MATLAB | The MathWorks, Inc | RRID:SCR_001622 | |
| Software, algorithm | SignalP v. 5.0 | DOI:10.1038/s41587-019-0036-z; *Almagro Armenteros et al., 2019* | RRID:SCR_015644 | |
| Software, algorithm | NetNGlyc - 1.0 | DOI:10.1142/9789812799623_0029; *Gupta and Brunak, 2001* | RRID:SCR_001570 | |
| Software, algorithm | big-PI Predictor | DOI:10.1006/jmbi.1999.3069; *Eisenhaber et al., 1999* | RRID:SCR_001599 | |
| Software, algorithm | TMHMM-2.0 | DOI:10.1006/jmbi.2000.4315; *Krogh et al., 2001* | RRID:SCR_014935 | |
| Software, algorithm | AlphaFold2 | DOI:10.1038/s41586-021-03819-2; *Jumper et al., 2021* | N/A | Version 2.3.2; https://alphafold.ebi.ac.uk/ |
| Software, algorithm | PyMOL | Schrödinger, Inc | RRID:SCR_000305 | |

