## [Editor Report]

This convincing and fundamental study uses unbiased genome-wide association analysis to identify a gene, called Bero, encoding Ly6/α-neurotoxin family protein, that affects the way larval *Drosophila* respond to nociceptive stimuli. This discovery is followed up by the identification of neurons in which Bero function is relevant for the modulation of nociceptive behaviour, namely the abdominal leucokinin-producing neurons. These neurons are activated by nociceptive sensory neurons and can initiate escape behavior. In these neurons Bero modulates both persistent and evoked activities. This elegant work will be of interest to neurobiologists working on genes, neural circuits, and behaviour.

---

## [Decision Letter]

**Decision letter after peer review:**

Thank you for submitting your article "The GPI-anchored Ly6 protein Belly roll regulates *Drosophila melanogaster* escape behaviours by modulating the excitability of nociceptive peptidergic interneurons" for consideration by *eLife*. Your article has been reviewed by 2 peer reviewers, and the evaluation has been overseen by a Reviewing Editor and K VijayRaghavan as the Senior Editor. The reviewers have opted to remain anonymous.

Essential revisions:

1) Please add a heterozygous effector-control (+/+, +/UAS-beroRNAi) in the relevant experiments, to make sure that phenotypes do not arise from leaky expression from the UAS transgene.

2) With regards to GCamp imaging analysis it is not clear why in Figure 4D, the ABLK neurons in the LK>bero RNAi, ATR (+) condition do not show lower spontaneous activity than the control retinal (+) condition at baseline. It might be helpful to show not only the averaged data and standard deviation but all recording traces in the figures as well. This could help clarify why the effects of deleting Bero in ABLK neurons depend on the presence or absence of trans retinal.

3) Discuss further alternative scenarios for unexpected or seemingly contradictory results. This pertains to the (seeming?) discrepancy between the results for silencing the ABLK neurons by kir versus their inhibition by GtACR1(Figure 5D-E), and for observing the very same effect, namely longer nociceptive-behaviour latency, for inhibiting and for driving the ABLK neurons (Figure 5E-F).

*Reviewer #1 (Recommendations for the authors):*

The conclusions of this paper are mostly well supported by the data, although some aspects of image acquisition and data analysis need to be clarified and extended.

1) Bero localization in the cell: Membrane localization was suggested by sequence information. Bero-YFP could be seen in the cell body strongly (and in other dendrites?), but not in the axon terminals. It would be helpful if the authors could provide more detailed confocal images with other markers to show the specific areas of Bero localization. High-resolution images in the cell body should be able to show whether Bero is localized in the membrane. It would be interesting to know whether its localization changed by stress conditions, etc.

2) GCamp imaging analysis: I am a bit confused why in Figure 4D, the ABLK neurons in the LK>bero RNAi, ATR (+) condition do not show lower spontaneous activity than the control retinal (+) condition at baseline. I would have expected that deltaF/F0 would be less than that in the control retinal (+) condition or close to 0, but the data in Figure 4D show that spontaneous activity is similar in both conditions. Why is this? It might be helpful to show not only the averaged data and standard deviation but all recording traces in the figures as well. This could help clarify why the effects of deleting bero in ABLK neurons depend on the presence or absence of trans retinal. It would also be interesting to know about the neural activity at axon terminals. Do you see a similar pattern of spontaneous firing?

*Reviewer #2 (Recommendations for the authors):*

I think the manuscript would improve substantially if it were possible to exclude the effects of leaky transgene expression.

For example, in Figure 2E the experimental group uses as a driver strain *nsyb*-Gal4 crossed to UAS-beroRNAi as the effector strain. This results in double-heterozygote offspring (*nsyb*-Gal4/+, +/UAS-beroRNAi). It is good and established practice that such experiments need two types of control:

One such canonical control is a heterozygous driver-control (*nsyb*-Gal4/+, +/+); such a driver-control the authors use in this Figure and throughout.

The second is a heterozygous effector-control (+/+, +/UAS-beroRNAi); this the authors do not seem to be used in this Figure, and indeed throughout. However, such effector controls are essential, in this Figure and throughout, to make sure that phenotypes do not arise from leaky expression from the UAS transgene.

As it stands, this leaves key experiments inconclusive, in this Figure and for related cases throughout.

In two cases, I think it would be important to discuss alternative scenarios for unexpected or seemingly contradictory results. This pertains to the (seeming?) discrepancy between the results for silencing the ABLK neurons by kir versus their inhibition by GtACR1 (possible that there is rebound activation for the latter? chloride spikes possible?) (Figure 5D-E), and for observing the very same effect, namely longer nociceptive-behaviour latency, for inhibiting and for driving the ABLK neurons (Figure 5E-F).

Given the striking phenotype of the bero-knockdown on 'spontaneous' rhythmic and persistent neuronal activity (Figure 4A), maybe there are discoveries to be made by observing undisturbed, free locomotion in these animals at a higher resolution.

P4: Maybe try to better motivate why you focus on the ABLK neurons, and why you mention ABLK versus LK neurons in line 64 versus 67.

P4: Better to say "Based on our results regarding the ABLK neurons…"?

P5, line 80: *w1118* is not a wild-type strain.

General readers might benefit from the authors' expertise and opinion on why their DGRP2.0 approach is more suitable than alternative approaches in the field (e.g. Long et al. Trends Genet 2014).

P5, line 91: You would still need to convince me that the four metrics you use indeed are sufficiently distinct for the GWA purpose.

Figure 1D and related text: The paper would benefit, I think, from a more explicit analysis/ discussion as to whether the performance of the strains as reflected by the three response classes co-vary and what it means for the present analyses if they do/ do not co-vary.

Figure 1F and related text: The paper would benefit from a more explicit analysis/ discussion of other SNPs in the bero gene (if any). What is the meaning of the color code above the stippled line?

P6, line 102: I think it is too early in the analysis to speak of causation.

P6, line 111: Better to briefly specify these tools?

I understand that it might be because of *eLife*´s policy, but I would prefer quite some of the supplemental material be shown in the body text, including in particular Figure 2-S2, 3-1.

P7, line 135: Better to give essential detail on the nature of the mutant here?

P8, line 154: Better to speak of "subsets" of EH, IPC, etc neurons?

P8, line 159: I was wondering whether one would need to consider expression also outside the nervous system (also see P15, line 306-7)?

P8, line 167: Here it would seem good to "justify" why there is no analysis of eg the dimm+, EH^+^, etc neurons.

Figure 3A, C: It should be good to be more explicit about the fact that some neurons are eg LK^+^, but not expressing bero. P9, line 172 you mention 7 pairs, but I see but 3 in Figure 3C.

P9, line 194: How did you make sure there is no (sic) all-trans-retinal in the food?

Figure 4E, p10, line 197: You mention that in the controls there is a rise in signal BEFORE the light stimulation. Could an alternative be that this is a sampling error, that it is being observed by chance, given that the data are from only 3 animals?

Figure 4F, left: One might fancy a trend for ATR leading to reduced signals in the controls. Would that be an expected trend? Would it warrant discussion or follow-up experimentation?

Figure 5A: Given the effect of driving the SELK neurons, I was wondering what the effect of beroRNAi in these neurons would be in experiments like the ones in Figure 3.

P18, line 389: Correct as "affect"!

---

## [Author Response]

Essential revisions:1) Please add a heterozygous effector-control (+/+, +/UAS-beroRNAi) in the relevant experiments, to make sure that phenotypes do not arise from leaky expression from the UAS transgene.

To address this point, we have re-examined the key experiments with heterozygous effector-controls. Since genomic background can affect the phenotype (Figure 1B and Figure 2E), and all driver strains used in this manuscript could have different genomic backgrounds, we did not simply repeat the experiments with additional heterozygous effector-controls. Instead, we compared the phenotype of the heterozygous effector-controls (e.g., y[1] w[1118]/y[1] v[1]; +/+; UAS-bero[shRNA#2]attP2/+) with their corresponding controls (e.g., y[1] w[1118]/y[1] v[1]; +/+; attP2/+), as they share the same genomic background within each pair. These additional groups of effector-control tests ensure that any observed phenotypes were specific, rather than any non-specific effects from UAS transgene leaky expression.

In summary, we have re-examined the main experiment and included the data from additional groups of effector-control tests for these key experiments (see the modified plots in Figure 2E, 3E and 3F of our revised manuscript). We have also included data from additional groups of effector-control tests in the other experiments (see Figure 5D–5G, Figure 6B, and Figure 6—figure supplement 1). The change has been described in lines 115–119, 201–203 and 303–306 of our revised manuscript.

2) With regards to GCamp imaging analysis it is not clear why in Figure 4D, the ABLK neurons in the LK>bero RNAi, ATR (+) condition do not show lower spontaneous activity than the control retinal (+) condition at baseline. It might be helpful to show not only the averaged data and standard deviation but all recording traces in the figures as well. This could help clarify why the effects of deleting bero in ABLK neurons depend on the presence or absence of trans retinal.

To address this issue, we have included the recording traces of calcium responses in control and *bero* knockdown larvae in Figure 4—figure supplement 1B of our revised manuscript. In ATR (+) condition, persistent fluctuations in ABLK neurons of *bero* knockdown animals appeared to be enhanced compared to controls. We postulate that the 561-nm laser utilized for jRCaMP1b excitation weakly activates ChR2.T159C in nociceptors, inducing weak nociceptive activity. Furthermore, *bero* knockdown ABLK neurons displayed increased sensitivity to nociceptive inputs than controls. As such, the baseline neural activity under ATR (+) conditions may not reflect proper spontaneous activity. Thus, we focused on the ATR (–) condition to compare the spontaneous activities. The observation has been described in lines 236–244 of our revised manuscript.

3) Discuss further alternative scenarios for unexpected or seemingly contradictory results. This pertains to the (seeming?) discrepancy between the results for silencing the ABLK neurons by kir versus their inhibition by GtACR1(Figure 5D-E), and for observing the very same effect, namely longer nociceptive-behaviour latency, for inhibiting and for driving the ABLK neurons (Figure 5E-F).

We have revised the Discussion section and incorporated additional scenarios to explain the unexpected results, which illustrate how the two distinct inhibition procedures caused the opposite effects on behavioural outputs (see lines 425–435 of our revised manuscript). We have also included scenarios to interpret how the CsChrimson-mediated activation of ABLK neurons results in the inhibition of nociceptive rolling response to thermal stimulation in the HPA experiment (see lines 408–424).

Reviewer #1 (Recommendations for the authors):The conclusions of this paper are mostly well supported by the data, although some aspects of image acquisition and data analysis need to be clarified and extended.1) Bero localization in the cell: Membrane localization was suggested by sequence information. Bero-YFP could be seen in the cell body strongly (and in other dendrites?), but not in the axon terminals. It would be helpful if the authors could provide more detailed confocal images with other markers to show the specific areas of Bero localization. High-resolution images in the cell body should be able to show whether Bero is localized in the membrane. It would be interesting to know whether its localization changed by stress conditions, etc.

In order to show the membrane localization and distribution of Bero proteins more clearly, we have included more detailed confocal images of Bero-YFP with the membrane marker hCD4-tdTomato in Figure 3C of our revised manuscript. We have also included a new panel showing the dendrite and axon terminal markers in ABLK neurons in Figure 3D of our revised manuscript. These images demonstrate that Bero localizes on the plasma membrane and distributes along both dendric and axonal regions in the neurons. The observation has been described in lines 187–196 of our revised manuscript. While it would be interesting to investigate whether its localization changes under stress conditions, this question should be explored in future studies.

2) GCamp imaging analysis: I am a bit confused why in Figure 4D, the ABLK neurons in the LK>bero RNAi, ATR (+) condition do not show lower spontaneous activity than the control retinal (+) condition at baseline. I would have expected that deltaF/F0 would be less than that in the control retinal (+) condition or close to 0, but the data in Figure 4D show that spontaneous activity is similar in both conditions. Why is this? It might be helpful to show not only the averaged data and standard deviation but all recording traces in the figures as well. This could help clarify why the effects of deleting bero in ABLK neurons depend on the presence or absence of trans retinal. It would also be interesting to know about the neural activity at axon terminals. Do you see a similar pattern of spontaneous firing?

We have addressed the main part of this issue in the ‘Essential revisions’ section ‘2’. In summary, the baseline neural activity under ATR (+) conditions may not reflect proper spontaneous activity. Thus, we focused on the ATR (–) condition for comparisons of the spontaneous activity. The observation has been described in lines 236–244 of our revised manuscript.

For the second question, it has been reported that the neural activity rise in the axon terminals of ABLK neurons occurs concurrently with that in the cell bodies (Okusawa et al., 2014). We also observed a similar pattern of spontaneous activity in neurites and cell bodies. To illustrate this, we have included a new graph depicting neural activities in the neurite regions in Figure 4—figure supplement 1A of our revised manuscript. The observation has been described in lines 212–216 of our revised manuscript.

Reviewer #2 (Recommendations for the authors):I think the manuscript would improve substantially if it were possible to exclude the effects of leaky transgene expression.For example, in Figure 2E the experimental group uses as a driver strain nsyb-Gal4 crossed to UAS-beroRNAi as the effector strain. This results in double-heterozygote offspring (nsyb-Gal4/+, +/UAS-beroRNAi). It is good and established practice that such experiments need two types of control:One such canonical control is a heterozygous driver-control (nsyb-Gal4/+, +/+); such a driver-control the authors use in this Figure and throughout.The second is a heterozygous effector-control (+/+, +/UAS-beroRNAi); this the authors do not seem to be used in this Figure, and indeed throughout. However, such effector controls are essential, in this Figure and throughout, to make sure that phenotypes do not arise from leaky expression from the UAS transgene.As it stands, this leaves key experiments inconclusive, in this Figure and for related cases throughout.

Please see our response to Essential revisions section 1.

In two cases, I think it would be important to discuss alternative scenarios for unexpected or seemingly contradictory results. This pertains to the (seeming?) discrepancy between the results for silencing the ABLK neurons by kir versus their inhibition by GtACR1 (possible that there is rebound activation for the latter? chloride spikes possible?) (Figure 5D-E), and for observing the very same effect, namely longer nociceptive-behaviour latency, for inhibiting and for driving the ABLK neurons (Figure 5E-F).

Please see our response to Essential revisions section 3.

Given the striking phenotype of the bero-knockdown on 'spontaneous' rhythmic and persistent neuronal activity (Figure 4A), maybe there are discoveries to be made by observing undisturbed, free locomotion in these animals at a higher resolution.

We have performed a new behavioural analysis on the undisturbed free locomotion in Control and *bero* RNAi animals at a higher resolution. The analysis results indicate that the larvae’s free locomotion is not affected by the 'spontaneous' rhythmic and persistent neuronal activity in ABLK neurons. We have included the tracking and analysis data in Figure 4—figure supplement 2 of our revised manuscript, and the observation has been described in lines 232–235.

P4: Maybe try to better motivate why you focus on the ABLK neurons, and why you mention ABLK versus LK neurons in line 64 versus 67.

We have included the reason why we focused on the ABLK neurons and corrected “Lk-neuron-specific *bero* knockdown” into “*bero* knockdown in ABLK neurons” in line 69–70 in our revised manuscript. In summary, our results indicate that among all the *bero*-expressing neurons, only the *bero*-knockdown in ABLK neurons caused enhanced rolling escape behaviour.

P4: Better to say "Based on our results regarding the ABLK neurons…"?

We have corrected “Based on our physiological analysis of ABLK neurons” into “Based on our results regarding the ABLK neurons” in line 72 in our revised manuscript.

P5, line 80: w1118 is not a wild-type strain.

We have changed “the two commonly used wild-type strains, *w1118* and Canton-S” into “the two commonly used strains, *w1118* and Canton-S” in line 82 in our revised manuscript.

General readers might benefit from the authors' expertise and opinion on why their DGRP2.0 approach is more suitable than alternative approaches in the field (e.g. Long et al. Trends Genet 2014).

We chose to use the DGRP2.0 approach as it is a more appropriate method for identifying which SNPs are responsible for the behavioural variations. While the *Drosophila* Synthetic Population Resource (DSPR) is available, this approach examines the influence of a local haplotype on a particular phenotype, making it difficult to pinpoint the specific molecular variants associated with the phenotype (Long et al., 2014). Moreover, while the DSPR provides some benefits for behavioural studies as a set of recombinant inbred lines derived from several founder strains, it may not encompass the full range of genetic variations found in natural populations. Thus, the DGRP2.0 approach was considered to be better for the analysis of behavioural variations due to its more comprehensive representation of genetic diversity. To clarify this situation, we have included a description and cited Long et al., 2014 in line 329–332 of our revised manuscript.

P5, line 91: You would still need to convince me that the four metrics you use indeed are sufficiently distinct for the GWA purpose.

As you may have already considered, these metrics can be mutually dependent to some extent. Instead, in the GWA analysis result, the candidate variants are only partially overlapped for each metric (please see Figure 1—Source data 2). To clarify this, we have included a description of this point in lines 94–98 in our revised manuscript.

Figure 1D and related text: The paper would benefit, I think, from a more explicit analysis/ discussion as to whether the performance of the strains as reflected by the three response classes co-vary and what it means for the present analyses if they do/ do not co-vary.

In Figure 1D and the following GWA analysis, we categorized the rolling probability into three response classes (rolling probability in 2 seconds, 5 seconds, and 10 seconds) based on previous studies (Onodera et al., 2017; Tracey et al., 2003). We assumed that the three classes are strongly related to the temporal responsiveness of the strains. To examine this explicitly, we applied a correlation analysis and found that the three response classes are highly correlated with one another. To illustrate this result, we have included the corresponding plots in the Figure 1—figure supplement 1A and described in lines 90–94 of our revised manuscript.

Figure 1F and related text: The paper would benefit from a more explicit analysis/ discussion of other SNPs in the bero gene (if any). What is the meaning of the color code above the stippled line?

Although there are many genetic variants (85 SNPs, 4 insertions and 7 deletions) in *CG9336* gene among the DGRP strains, the GWA analysis did not yield any candidate variants other than the SNP (2L_20859945_SNP) (see Figure 1—source data 2). In Figure 1F and Figure 1—figure supplement 1B, the color of individual genetic variants (dots) indicates the respective chromosome to which they belong. We have included a description of this point in the corresponding figure legends of our revised manuscript.

P6, line 102: I think it is too early in the analysis to speak of causation.

We have changed ‘caused’ into ‘resulted in’ in line 109 in our revised manuscript.

P6, line 111: Better to briefly specify these tools?

We have included detailed information about these tools in the Materials and methods section in lines 691–700 of our revised manuscript.

I understand that it might be because of eLife´s policy, but I would prefer quite some of the supplemental material be shown in the body text, including in particular Figure 2-S2, 3-1.

We appreciate the reviewer’s suggestions regarding the presentation of supplemental materials. However, due to limited space constraints, we have included these materials as supplementary figures to keep the main body text focused and concise.

P7, line 135: Better to give essential detail on the nature of the mutant here?

We have provided the essential detail on the nature of the mutant in lines 147–150 in our revised manuscript.

P8, line 154: Better to speak of "subsets" of EH, IPC, etc neurons?

We have added ‘a subset of’ in lines 167–168 before EH neurons, because only a subset of *Eh*-positive neurons express *bero*. On the other hand, all of IPC and ABLK do express *bero*, so we did not modify them.

P8, line 159: I was wondering whether one would need to consider expression also outside the nervous system (also see P15, line 306-7)?

As you may have considered, *bero* expressed in non-neuronal tissue could also contribute to nociception. Therefore, we have included a description of this possibility in lines 173–174 of our revised manuscript.

P8, line 167: Here it would seem good to "justify" why there is no analysis of eg the dimm+, EH^+^, etc neurons.

We have evaluated the nociceptive function of *bero* in EH and IPC neurons and included the results in Figure 3E in our revised manuscript. The observation has been described in lines 182–184.

Figure 3A, C: It should be good to be more explicit about the fact that some neurons are eg LK^+^, but not expressing bero. P9, line 172 you mention 7 pairs, but I see but 3 in Figure 3C.

We have replaced Figure 3C with additional confocal images and included individual optical sections depicting ABLK neurons with high resolution, which clearly show that all ABLK neurons (14 neurons in total) express Bero-YFP.

P9, line 194: How did you make sure there is no (sic) all-trans-retinal in the food?

We have not validated that the food was free from ATR. Therefore, we have replaced ‘lacking’ with ‘with no additional’ in lines 219–220 and 222 of our revised manuscript. For practical purposes, the current food condition appears to be appropriate, as animals in the ATR (–) groups hardly showed any response.

Figure 4E, p10, line 197: You mention that in the controls there is a rise in signal BEFORE the light stimulation. Could an alternative be that this is a sampling error, that it is being observed by chance, given that the data are from only 3 animals?

We appreciate the reviewer’s comment. We agree that the rise in the control signal prior to light stimulation in Figure 4E may have happened by coincidence, as similar rises are consistently observed during observations irrespective of the stimulation.

Figure 4F, left: One might fancy a trend for ATR leading to reduced signals in the controls. Would that be an expected trend? Would it warrant discussion or follow-up experimentation?

We appreciate the reviewer’s inquiry. Although the maximum amplitude of these activities was slightly lower in the ATR (+) control compared to the ATR (–) control, the difference is not statistically significant (*P* = 0.0646). Therefore, we will not be discussing or conducting follow-up experimentation on this point in our revised manuscript.

Figure 5A: Given the effect of driving the SELK neurons, I was wondering what the effect of beroRNAi in these neurons would be in experiments like the ones in Figure 3.

As shown in Figure 3A and 3C, *bero* is not expressed in SELK neurons, which is explicitly described in our revised manuscript. Therefore, we assume that *bero* RNAi in SELK would not have had any effect on the rolling behaviour.

P18, line 389: Correct as "affect"!

We have corrected ‘effect’ as ‘affect’ in line 445 in our revised manuscript.